

Regional Assessment and Uncertainty Analysis of Carbon and Nitrogen Balances at
cropland scale using the ecosystem model LandscapeDNDC
Odysseas Sifounakis[1], Edwin Haas[2], Klaus Butterbach-Bahl[2,3] and Maria P. Papadopoulou[1]
[1] Laboratory of Physical Geography and Environmental Impacts, School of Rural, Surveying
and Geoinformatics Engineering, National Technical University of Athens, Athens, 15780,
Greece
[2] Institute of Meteorology and Climate Research (IMK-IFU), Karlsruhe Institute of Technology,
Kreuzeckbahnstr. 19, D-82467 Garmisch-Partenkirchen, Germany
[3] Department of Agroecology - Center for Landscape Research in Sustainable Agricultural
Futures - Land-CRAFT, Aarhus University, Aarhus, 8000, Denmark
*Correspondence to*: Edwin Haas (edwin.haas@kit.edu)





Abstract
The assessment of cropland carbon and nitrogen (C & N) balances play a key role to identify
cost effective mitigation measures to combat climate change and reduce environmental
pollution. In this paper, a biogeochemical modelling approach is adopted to assess all C & N
fluxes in a regional cropland ecosystem of Thessaly, Greece. Additionally, the estimation and
quantification of the modelling uncertainty in the regional inventory are realized through the
propagation of parameter distributions through the model leading to result distributions for
modelling estimations. The model was applied on a regional dataset of approximately 1000
polygons deploying model initializations and crop rotations for the 5 major crop cultivations
and for a timespan of 8 years. The full statistical analysis on modelling results yields for the C
balance carbon input fluxes into the soil of $12.4 \pm 1.4$ tons C ha$^{-1}$ yr$^{-1}$ and output fluxes of 11.9
$\pm 1.3$ tons C ha$^{-1}$ yr$^{-1}$, with a resulting average carbon sequestration of $0.5 \pm 0.3$ tons C ha$^{-1}$ yr$^{-1}$
$^{1}$. The averaged N influx was $212.3 \pm 9.1$ kg N ha$^{-1}$ yr$^{-1}$ while outfluxes were estimated on
average of $198.3 \pm 11.2$ kg N ha$^{-1}$ yr$^{-1}$. The net N accumulation into the soil nitrogen pools was
estimated to $14.0 \pm 2.1$ kg N ha$^{-1}$ yr$^{-1}$. The N outflux consist of gaseous N fluxes composed by
$N_2O$ emissions $2.6 \pm 0.8$ kg $N_2O$-N ha$^{-1}$ yr$^{-1}$, NO emissions of $3.2 \pm 1.5$ kg NO-N ha$^{-1}$ yr$^{-1}$, $N_2$
emissions $15.5 \pm 7.0$ kg $N_2$-N ha$^{-1}$ yr$^{-1}$ and $NH_3$ emissions of $34.0 \pm 6.7$ kg $NH_3$-N ha$^{-1}$ yr$^{-1}$, as
well as aquatic N fluxes (only nitrate leaching into surface waters) of $14.1 \pm 4.5$ kg $NO_3$-N ha$^{-1}$
yr$^{-1}$, N fluxes of N removed from the fields in yields, straw and feed of $128.8 \pm 8.5$ kg N ha$^{-1}$ yr$^{-1}$
$^{1}$.
KEYWORDS: climate change mitigation, greenhouse emissions, ecosystem modelling,
cropland carbon and nitrogen balance, inventory, Thessaly region, LandscapeDNDC





Graphical abstract: Result distributions of all nitrogen fluxes with means and medians

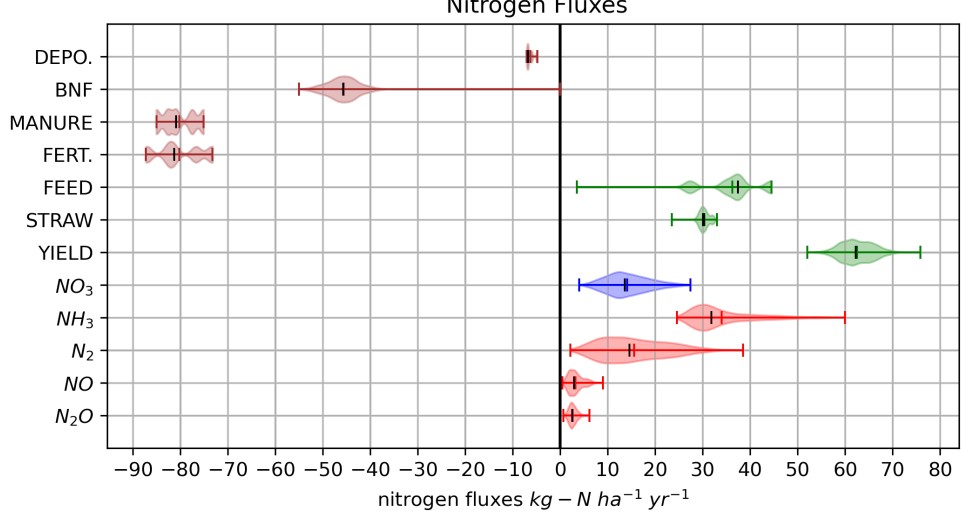



## 1 Introduction

Food security as well as the agricultural productivity depend to a major extend on the applied
nitrogen (N) fertilizers (Klatt et al., 2015a). Worldwide, the N fertilizer use for the years 1960 to
2005 has increased from 30 to 154 million tons (IFADATA, 2015). In Europe, the increase of
yields in arable land and grassland systems was 45-70% since 1950 (EFMA, 2009) due to the
agricultural production systems intensification. Excessive use of N fertilizers, though
beneficially affecting the yield, could cause a harmful impact to the environment, e.g. increased
gaseous emissions and aquatic fluxes of nitrous oxide ($N_2O$) to the atmosphere and leaching
of nitrate ($NO_3$) into water bodies (Erisman et al., 2011; Galloway et al., 2013; Kim et al., 2015)
The $N_2O$ poses a twofold environmental threat. From the one hand, it is a strong greenhouse
gas with a warming potential of 300 times greater (in a 100-year time period) than carbon
dioxide ($CO_2$) and from the other hand, it is a major driver of ozone depletion in stratosphere
(Ravishankara et al., 2009). The fertilizer use aiming at the increase of the agricultural
production is the most crucial anthropogenic source of atmospheric $N_2O$, which at present
contributes for approximately 45% of total anthropogenic $N_2O$ emissions on a global scale
(Jones et al., 2014). Because of the global population growth and thus a growing food and
feed demand (Godfray et al., 2010), the fertilizer use will probably increase. Consequently, the
prediction of the current business-as-usual scenarios show doubled anthropogenic $N_2O$
emissions by the year 2050 (Davidson and Kanter, 2014). The European countries have
recently set up bilateral agreements in order to reduce $N_2O$ emissions from cultivated crop
lands (EU-Commission, 2014). Similarly, the European Nitrates Directive (EU-Commission,
2019; Musacchio et al., 2020) aims at $NO_3$ leaching reduction to water bodies to avoid both an
increase of eutrophication (Camargo and Alonso, 2006) and drinking water pollution. Because
of the hazardous $N_2O$ and $NO_3$ effects, agricultural systems are necessary to be evaluated for
their profitability and productivity as well as for their impacts to the environment.
The $N_2O$ and $NO_3$ production and consumption in agricultural lands are regulated to a large
extend by N plant uptake and, also, the microbial processes of denitrification and nitrification
(Butterbach-Bahl et al., 2013). The factors controlling both the microbial metabolism and plant





N uptake are a) soil conditions (Butterbach-Bahl et al., 2013) and b) cultivation management
practices e.g. crop rotation, fertilizing amount and timing, and ploughing (Smith et al., 2008).
In order to reach a minimization of the environmental footprint of feed and agricultural
production while securing the global food security (Garnett et al., 2013), it is mandatory to
tighten the N cycling on intensified agricultural systems e.g., by harmonizing N demand of
crops with soil N availability driven by fertilization.
A number of environmental/ecosystem models have been developed and used to describe the
structure of multiple biogeochemical processes (Wainwright and Mulligan, 2004.). For the
estimation/quantification of the GHGs emissions from different agroecosystems, modelling
approaches are constantly gaining ground due to the in-situ data limitation (field campaign and
laboratory costs) and the variation in spatial and temporal scales. The simulated results may,
also, have uncertainties resulting from different sources, which can be, though, quantified
increasing the accuracy of the estimates. Mechanistic models integrating relevant processes,
which simulate agricultural production, and, also, reactive N losses to the environment are
valuable tools to infer practices for a sustainable agriculture. In recent years, process-based
biogeochemical models such as e.g. DNDC (Li, 2000), DAYCENT (Parton et al., 1998),
ECOSSE (Bell et al., 2012) and CERES-EGC (Gabrielle et al., 2006) have proven their
applicability to simulate $N_2O$ emissions and $NO_3$ leaching from various land uses. Despite the
fact that their accuracy is being assessed against in-situ data, few studies are reported to use
sensitivity and uncertainty analyses in total N and C cycling simulation by process-based
models (Verbeeck et al., 2006).
In this analysis, the process-based bio-geochemical model LandscapeDNDC (Haas et al.,
2013) was applied to the agricultural cropland systems in the region of Thessaly (Greece). The
objective of our study was to i) assess and report the cropland C and N balance including all
associated fluxes such as e.g. $CO_2$, $N_2O$ and $NH_3$ emissions, $NO_3$ leaching as well as the soil
carbon stock changes; ii) to assess and quantify the modelling uncertainty of the simulated C
and N balance and flux estimations as requested by the IPCC (IPCC, 2019); and iii) to
demonstrate the feasibility and robustness of a regional uncertainty assessment methodology



for C and N cycling by propagating 500 joint parameter and input data distributions through
the model (each representing a full regional inventory simulation) yielding regional result
distributions for any modelling estimations.

2 Material and Methods
2.1 Model description
LandscapeDNDC is a modular process-based ecosystem model for simulating the bio-
geochemical change of C and N in croplands, forest and grassland systems at both site and
regional scale. The modules combined are about plant growth, micro-meteorology, water
cycling, physico-chemical-plant and microbial C and N cycling and exchange processes with
atmosphere and hydrosphere of terrestrial ecosystems. LandscapeDNDC is a generality of the
plant development and soil biogeochemistry of the agricultural DNDC and Forest-DNDC (Li,
2000). There is a successful application of earlier model versions in a number of studies, e.g.
water balance (Grote et al., 2009; Holst et al., 2010), plant growth (Cameron et al., 2013;
Werner et al., 2012), $NO_3$ leaching (Kim et al., 2015; Thomas et al., 2016) and soil respiration
and gas emission trace (Chirinda et al., 2011; Kraus et al., 2014; Molina-Herrera et al., 2015).
For the initialization of LandscapeDNDC physical and chemical site-specific soil profile
information is used (specified for different soil depths): Soil organic carbon (SOC) and nitrogen
(SON) content, soil texture (clay, sand and silt content), of the plant growth and soil
biogeochemistry, bulk density, pH value, saturated hydraulic conductivity, field capacity and
wilting point. Daily or hourly climate data of air temperature (max, min and average), N
deposition, precipitation, and atmospheric $CO_2$ concentration are used in LandscapeDNDC in
combination with agricultural management practices e.g. crop planting and harvesting,
fertilizing (synthetic and organic) or feed cutting and tilling are used to drive LandscapeDNDC
simulations. Regarding fertilization management three types of mineral fertilizers, i.e. urea,
compound fertilizers based on $NH_4$ and $NO_3$ as well as organic amendments, i.e. green
manure, farmyard manure, slurry, straw, bean cake and compost are currently considered.



The growth of crops and grasses is similar to the DNDC approach using two major parameters
that describe seasonal plant development (cumulative temperature degrees days) and
maximum reachable biomass under optimum conditions (Li, 2000) while daily growth
limitations due to water and nutrient availability are considered. Model parameters describing
soil and vegetation characteristics are obtained from an external parameter library. In
LandscapeDNDC, the parameterization of the main cultivated commodity crops in Europe
occurs by default parameter sets representing an average plant type while process parameter
values for micro-meteorology, water cycle and bio-geochemical processes were obtained from
previous validation studies, e.g. (Klatt et al., 2015a; Molina-Herrera et al., 2016; Rahn et al.,
2012) proving that the LandscapeDNDC model could be universally applicable for similar
conditions.
For all simulations in the current study, site-specific crop parameterizations were derived in a
preceding analysis of various site scale simulations and validations of yield characteristics
across the region. An overview of the crops cultivated at the different study sites and detailed
information on specific crop rotations used to simulate crop growth are provided in Table A2
(supplementary material).
2.2   Case study description and input data
The region of Thessaly is located in Central Greece covering a total area of 14 000 $Km^2$, where
5000 $Km^2$ is lowland and approx. 2300 $Km^2$ and 6500 $Km^2$ are semi-mountainous and
mountainous land respectively. The plain of Thessaly is considered to be among the largest
agricultural land of the country (Kalivas et al., 2001) accounting for almost 410 000 ha, of which
about 370 000 ha is arable land where almost 80% is covered by annual and 10% by perennial
crops (ELSTAT, 2012). The crop/plant production of the region is around 14.2% (ELSTAT,
2012) of the total production of the country ($2^{nd}$ in Greece).
Soil input data for the region was available from the European Project Nitro Europe IP (Sutton
et al., 2013) based on the European Soil Database (ESDB v2.0, 2004) containing, soil type
and soil profile description of bulk density, SOC content, texture (sand, silt clay), pH value,



stone fraction, saturated hydraulic conductivity, wilting point and water-holding capacity in
various soil strata (Cameron et al., 2013). A regional soil dataset for the area of interest
contained about 1500 spatial polygons out of which approximately 1000 covered the cultivated
cropland that was finally simulated. The climate data for the regional simulations was derived
at polygon level from gridded ERA5 climate data for Greece.
2.3    Agricultural Management and model input data processing
The total cultivated area and the respective yields for the years 2010 to 2016, used in the
current analysis were obtained from the Hellenic Statistical Authority (ELSTAT). Moreover,
data associated with the animal capital for the respective years was also provided (ELSTAT)
in order to estimate the annual manure production distributed in the region however no data is
available on whether and how much of the manure is used in croplands. For the water
management, the percentage of irrigated and non-irrigated land (estimated to almost 50% for
each case) was also given (ELSTAT) while indicative sets of irrigation management data were
acquired through the River Basin Management Plans of the Special Secretariat for Water,
Ministry of Environment and Energy (YPEKA, Portmann et al., 2010). The irrigation water
volumes were estimated based on the crops needs and the minimum and maximum quantities
necessary according to literature while using upscaling tools to get the regional values. The
fertilization data sets were provided by Fertilizer Producers and Merchandiser Association
(FPMA) for the recent years (2010-2016) and are equated to the annual consumed quantities
on a national level, scaled down to a regional level based on crop pattern in the Region of
Thessaly cultivated land.
In this study, the five main crops maize, wheat, clover, cotton and barley were considered,
covering the majority of the cultivated arable land in the region (over 95%) while the remaining
cropland was included acquiring the final corrected land/crop coverage. In Table 1 the resulting
crop rotation scenarios (R1 - R5) are presented for the evaluation period 2012 - 2016. Note,
each rotation sequence (R1 – R5) is shifted in time such that for each year, each crop appears
exactly in one rotation. Based on the crop cover contribution in each simulated year the crop





rotation contribution factors were estimated and are summarized in Table 2. The management
practices were based on the general agricultural practices applied in the region and information
provided by farmers.

*Table 1. Summary of the crop rotation scenarios (R1- R5) for the region of Thessaly. The crop abbreviations corn,*
*wiwh, clover, cott and wbar refer to maize (food corn and silage maize), winter wheat, clover (legume feed crops*
*s.a. alfalfa or vetch), cotton and winter barley respectively.*

| year | R1 | R2 | R3 | R4 | R5 |
|---|---|---|---|---|---|
| 2012 | clover | cotton | wbar | corn | wiwh |
| 2013 | cotton | wbar | corn | wiwh | clover |
| 2014 | wbar | corn | wiwh | clover | cotton |
| 2015 | corn | wiwh | clover | cotton | wbar |
| 2016 | wiwh | clover | cotton | wbar | corn |


*Table 2. Crop cultivation area contribution per year to the aggregation of the five rotations; data constant across*
*the region of Thessaly*

| Crop Rotation Contribution [% / 100] | | | | | |
|---|---|---|---|---|---|
| Years | R1 | R2 | R3 | R4 | R5 |
| 2012 | 0.15 | 0.15 | 0.45 | 0.11 | 0.14 |
| 2013 | 0.13 | 0.29 | 0.09 | 0.10 | 0.39 |
| 2014 | 0.29 | 0.13 | 0.10 | 0.35 | 0.12 |
| 2015 | 0.15 | 0.11 | 0.43 | 0.16 | 0.16 |
| 2016 | 0.10 | 0.36 | 0.14 | 0.14 | 0.25 |



2.4   Uncertainty analysis
As stated in the IPCC 2006 guidelines and updated in 2019, the assessment of uncertainty is
considered a major and crucial/mandatory component when compiling regional or national
GHG emission inventories (Larocque et al., 2008). The difference in scale in which the model
is used results in divergent errors of the C and N dynamics prediction across different climate
zones and scales. Thus, uncertainty analysis is a crucial step towards a higher quality decision





making process. The sources of uncertainty can vary and are related to a) the initial conditions
(starting values), b) the drivers (e.g. climate and crop management data), c) the conceptual
model uncertainty and d) the parameter uncertainty of the various processes (Refsgaard et al.,
2007; Wang and Chen, 2012).
Santarbarbara (2019) performed a Bayesian Model Calibration and Uncertainty Analysis using
a Monte Carlo Markov Chain (MCMC) approach targeting uncertainties associated to the data
(bulk density, SOC, pH, clay content) of the initial soil conditions, drivers (cropland
management such as fertilization/manure rates & timing, harvest & seeding timing, tillage
timing) and bio-geochemical process parameterizations.
In order to identify the most sensitive process parameters with a reduced number of model
simulations, the Morris method (Morris, 1991) obtains a hierarchy of parameters influence on
a given output (gaseous N fluxes) and evaluates whether a non-linearity exists or not. (Morris,
1991) proposed that this order can be assessed through the statistical analysis of the changes
in the model output, produced by the "one-step-at-a-time" changes in "n" number of proposed
parameters. Incremental steps of each parameter range, lead to identifying which ones have
substantial influences over the concerned results, without neglecting that some effects could
cancel each other (Saltelli et al., 2000), leading to the identification of the 24 most sensitive
process parameters (Houska et al., 2017; Myrgiotis et al., 2018b).

2.4.1 Metropolis – Hastings algorithm
The Markov Chain Monte Carlo (MCMC) Metropolis–Hastings algorithm results in numerous
parameter sets that approximate the posterior joint parameter distribution by performing a
random walk through the space of joint parameter values. This probability evaluation of the
data obtained from each step leads to the update of the initial uniform parameter distributions.
Bayes' formula relating conditional probabilities may become a powerful and practical
computational tool when combined with Markov chain processes and Monte Carlo methods,
so-called Markov Chain Monte Carlo (MCMC). A Markov chain is a special type of discrete



stochastic processes wherein the probability of an event depends only on the event that
immediately precedes it. Integrating parameters (θ) and observation data (D) into Bayes' rule
results in the formula:

$$P(\theta|D) = \frac{P(D \mid \theta) * P(\theta)}{P(D)} \qquad \text{2.1}$$

where $P(D \mid \theta)$, the probability of the data, is used to obtain the probability of these parameters
updated by the data: $P(\theta|D)$ where the evidence is computed as:

$$P(D) = \int likelyhood \cdot prior \, \cdot \, d\theta \qquad \text{2.2}$$

where $P(D)$ can be numerically approximated with the aforementioned MCMC method (Robert
and Casella, 2011).
The method uses prior knowledge concerning the sources of the model uncertainty to obtain
a narrowed posterior distribution for each one of the sources. By propagating the parameter
distributions through the model, the overall uncertainty in the model results can be quantified.
In the current analysis, 500 joint parameter sets were sampled from the posterior distributions
in combination with input data perturbations as reported by Santabarbara (2019) and were
deployed in simulations (propagation through the model) for the regional inventory leading to
500 inventory simulations. A statistical analysis was, afterwards, applied to estimate the
updated regional and temporal result distributions.

2.5   Statistical methods and data aggregation
2.5.1   Regional result aggregation
One full regional inventory simulation consists of 10 individual inventory simulations: Five (5)
different crop rotations for irrigated and rain feed conditions were simulated in parallel (see
section 2.3). The results of the crop rotations were aggregated according to the crop shares





per year (see Table 2) accounting for all effects of the different crops cultivated in the region
for irrigated and rain feed conditions. The final inventory simulation results were obtained by
considering irrigated versus rain feed water management. The final inventory contains
simulation results aggregated to area weighted yearly means across the total simulation
domain accounting for the cropland area of each polygon.

2.5.2   Uncertainty quantification and statistical analysis
A regional aggregation was performed for all 500 uncertainty simulations. All the uncertainty
results were finally reported via statistical measures evaluating the 500 regional uncertainty
simulation runs reporting mean values, standard deviation, medians and the 25 and 75
interquartile ranges (IQR, Q25 to Q75).

3    Results Analysis and Evaluation
The simulation time span was from 2009 to 2016, while the years 2009 – 2011 were used as
spin-up to get all soil C and N pools into equilibrium after the initialization. Therefore, reported
simulation results are limited to years 2012 - 2016. The assessment of the regional C and N
balances (CB and NB) were obtained - as a consequence of the uncertainty quantification -
resulting in distributions and therefore reported by statistical measures such as mean/median
or interquartile ranges of the uncertainty ensemble.

3.1    Regional yield simulations and validation
The evaluation of the model performance in estimating the NB and CB components was
analyzed based on the comparison of the simulated yield values with the observed yield data
provided by the Hellenic Statistical Authority (ELSTAT), averaged for the total simulated
period.



### 3.1.1 Crop yields and feed production

For model validation, datasets of crop yields from Hellenic Statistical Authority (ELSTAT) were used. Table 3 summarizes the aggregated regional crop yields for all the simulated years and the respective mean, median and standard deviation values resulted from the statistical analysis of the simulation results together with the observed yield and feed production provided by the Hellenic Statistical Authority (ELSTAT). Simulated yields consist for cotton of the cotton bolls, clover feed is the total cutting and harvested above ground biomass, for wheat and barley is the grain yield and for maize is accounted grain ear and the stems. Based on the observations, maize appears to be the dominant crop with an average yield of 12 tons ha$^{-1}$, followed by clover product of 8.4 tons ha$^{-1}$. The rest of the three crop yields appear to be in the same order of magnitude from 3.3 up to 3.4 tons ha$^{-1}$.

*Table 3. Simulated and observed yields and feed production [tons dry matter ha$^{-1}$] in the region of Thessaly. All results are based on statistical aggregation across all polygons, rotations, years and finally across all 500 UA inventory simulations. The observed values of dry matter (DM) are provided by the Hellenic Statistical Authority.*

| | Simulated crop yield and feed distributions [tons dry matter ha$^{-1}$] | | | Observed [tons dry matter ha$^{-1}$] |
|---|---|---|---|---|
| Crops | Median | Mean | standard deviation | Mean |
| Cotton | 3.5 | 3.3 | 0.8 | 3.3 |
| Clover | 9.8 | 9.6 | 0.6 | 8.4 |
| Wheat | 3.9 | 3.6 | 0.9 | 3.4 |
| Barley | 4.7 | 4.5 | 1.0 | 3.3 |
| Maize[1] | 10.2 | 9.9 | 1.4 | 12.0 |

[1] *Observation data for maize did not distinguish between food corn and silage maize.*

Additionally, the simulated average yield of cotton was estimated to 3.3 ± 0.8 tons DM ha$^{-1}$, wheat to 3.6 ± 0.9 tons DM ha$^{-1}$, barley 4.5 ± 1 tons DM ha$^{-1}$, maize 9.9 ± 1.4 tons DM ha$^{-1}$. As for the feed, the clover was estimated to 9.6 ± 0.6 tons DM ha$^{-1}$. The average nitrogen use efficiency (NUE) across time and space is 63.29%.



Figure 1 presents the uncertainties of the simulated crop yield across the whole evaluation
time span 2012 -2016 both in irrigated and rain feed conditions. As shown, corn shows a much
more narrow distribution with a higher median for the irrigated scenario compared to the rain
feed while shows the same extreme value variations. To the contrary, winter barley has a wider
distribution and slightly higher median for the irrigated scenario and, also, a wider extreme
value variation. As for cotton, the distribution appears to be bimodal for the rain feed scenario
in which the median is also lower than the one in the irrigated case. In addition, the extreme
value variation is wider in the latter case. Finally, for the example of winter wheat irrigated and
rain feed scenarios reach the same results.

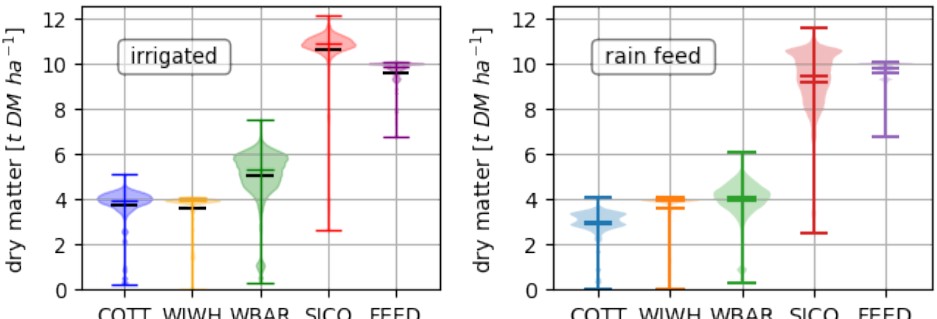


*Figure 1. Simulated crop yield uncertainties across the evaluation time span 2012 - 2016 for irrigated and rain feed*
*conditions. Horizontal lines indicate median, mean, maximum and minimum values of the distributions.*

## 3.2    Regional Carbon and Nitrogen Balance
### 3.2.1    Carbon Balance (CB)
For the CB, Figure 2 presents average C input fluxes into the soil of 12.4 $\pm$ 1.4 tons C ha$^{-1}$ yr$^{-1}$
$^{1}$ (with inter quartile ranges (IQR) from Q25 to Q75 of 12.1 to 13.2 tons C ha$^{-1}$ yr$^{-1}$) and output
fluxes of 11.9 $\pm$ 1.3 tons C ha$^{-1}$ yr$^{-1}$ with IQR from 11.6 to 12.7 tons C ha$^{-1}$ yr$^{-1}$. The resulting
carbon sequestration was estimated to 0.5 $\pm$ 0.3 tons C ha$^{-1}$ yr$^{-1}$ with IQR from 0.4 to 0.7 tons
C ha$^{-1}$ yr$^{-1}$ (data summarized in Table 4).




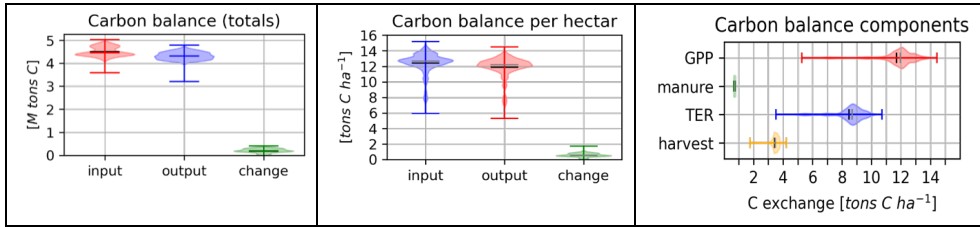

*Figure 2. Carbon balance for cropland cultivation for the region of Thessaly: a) Total carbon balance of cropland*
*soils in mio. tons C, b) averaged Carbon Balance in tons C ha$^{-1}$ and c) averaged fluxes across the region and the*
*years 2012-2016. (Positive change equals soil C sequestration).*

The input fluxes consist of annual gross primary productivity (GPP) of $11.7 \pm 1.4$ tons C ha$^{-1}$
yr$^{-1}$ with IQR from 11.4 to 12.4 tons C ha$^{-1}$ yr$^{-1}$ and carbon applied to soils in manure estimated
by $0.7 \pm 0.001$ tons C ha$^{-1}$ yr$^{-1}$ (see Table 4). This compares on the other hand to respirative
carbon fluxes from the soil to the atmosphere (TER) of $8.5 \pm 1.1$ tons C ha$^{-1}$ yr$^{-1}$ with IQR from
8.2 to 9.1 tons C ha$^{-1}$ yr$^{-1}$ and carbon fluxes via exported crop yields and feed (including all
straws and removed crop residues) of $3.4 \pm 0.3$ tons C ha$^{-1}$ yr$^{-1}$ with IQR from 3.4 to 3.6 tons
C ha$^{-1}$ yr$^{-1}$. The aggregation of the carbon fluxes to the regional level of approx. 360 000 ha of
cropland results in $4.25 \pm 0.20$ M tons C yr$^{-1}$ by GPP, $0.25 \pm 0.01$ M tons C yr$^{-1}$ carbon influx
via organic fertilizers compared to $3.08 \pm 2.97$ M t C yr$^{-1}$ TER and $1.24 \pm 0.05$ M t C yr$^{-1}$ carbon
exports via crop yields and feed production leading to a net carbon sequestration of $0.5 \pm 0.3$
M tons C ha$^{-1}$ yr$^{-1}$ with IQR from 0.4 to 0.7 M tons C ha$^{-1}$ yr$^{-1}$ (M tons C as Million tons carbon).

*Table 4.* **Carbon Balance** *(per hectare) Assessment and Uncertainty Analysis of the of cropland cultivation at the*
*region of Thessaly, Greece. [1] mean; [2] standard deviation; [3] median; Interquartile ranges: [4] Q25: 25 quartile, [5]*
*Q75: 75 quartile are applied across the 500 values for the quantities in this table; [6] C-Inputs as the sum of the*
*absolute values of all the input fluxes of the 500 simulations; [7] C-Outputs as the sum of the absolute values of all*
*the output fluxes of the 500 simulations; [8] SOC-changes as the difference between the input and output fluxes of*
*each of the 500 simulations.*

|  | Mean[1] | Std[2] | Median[3] | Q25[4] | Q75[5] |
|---|---|---|---|---|---|
|  | [tons C ha$^{-1}$ yr$^{-1}$] | [tons C ha$^{-1}$ yr$^{-1}$] | [tons C ha$^{-1}$ yr$^{-1}$] | [tons C ha$^{-1}$ yr$^{-1}$] | [tons C ha$^{-1}$ yr$^{-1}$] |
|  |  |  |  |  |  |
| C-Inputs[6] | 12.4 | 1.4 | 12.7 | 12.1 | 13.2 |





| C-Outputs[7] | 11.9 | 1.3 | 12.2 | 11.6 | 12.7 |
|---|---|---|---|---|---|
| SOC-changes[8] | 0.5 | 0.3 | 0.5 | 0.4 | 0.7 |
| | | | | | |
| Input fluxes | | | | | |
| GPP | 11.7 | 1.4 | 12.0 | 11.4 | 12.4 |
| C in manure | 0.7 | 0.0 | 0.7 | 0.7 | 0.7 |
| | | | | | |
| Output fluxes | | | | | |
| TER | 8.5 | 1.1 | 8.7 | 8.2 | 9.1 |
| Biomass export | 3.4 | 0.3 | 3.5 | 3.4 | 3.6 |
| | | | | | |


3.2.2   Nitrogen balance (NB)
In

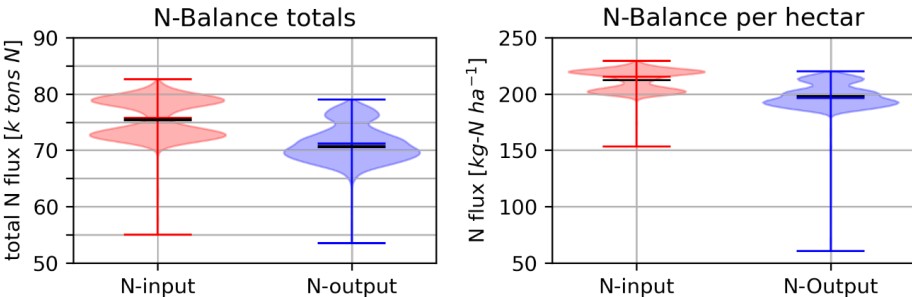


Figure 3 the assessment of the distribution of the NB with the in- and out-fluxes is presented.
The averaged nitrogen influx (represented by the uncertainty ensemble mean) per hectare was
estimated to 212.3 $\pm$ 9.1 kg N ha$^{-1}$ yr$^{-1}$ with IQR from 203.3 to 220.0 kg N ha$^{-1}$ yr$^{-1}$ while nitrogen
out-fluxes were estimated in average to 198.3 $\pm$ 11.2 kg N ha$^{-1}$ yr$^{-1}$ with IQR from 191.4 to
204.0 kg N ha$^{-1}$ yr$^{-1}$ (



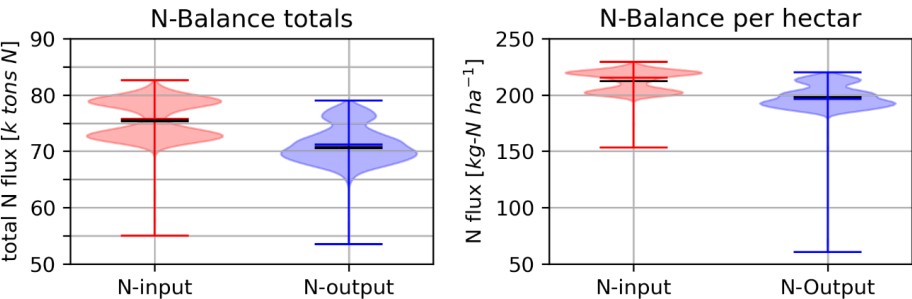


Figure 3) leading to a net N accumulation in the soil of 14.0 ± 2.1 kg N ha⁻¹ yr⁻¹ with IQR from


11.9 to 16.0 kg N ha⁻¹ yr¹.

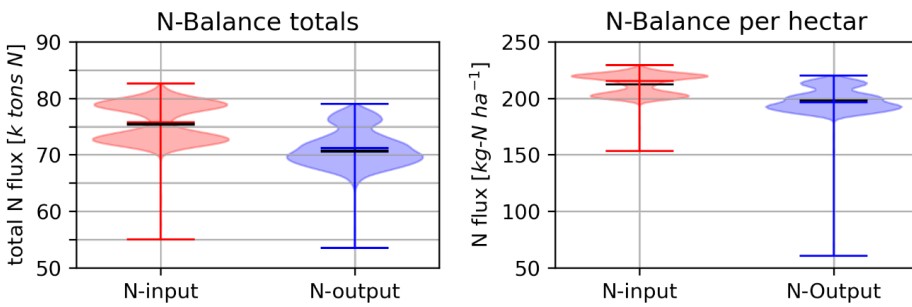


*Figure 3. Nitrogen balance for cropland cultivation for the region of Thessaly; a) Total NB in k-tons N and b)*
*averaged NB in kg N ha⁻¹; Data averaged for the years 2012-2016. Horizontal lines indicate mean (red), median*
*and minimum and maximum of the distribution.*

*Table 5. Nitrogen Balance (per hectar). Summary of the Assessment and Uncertainty Analysis of the **NB Fluxes***
*(per hectare) of cropland cultivation of the region of Thessaly, Greece. [1] N-Inputs as the sum of the absolute values*
*of all input fluxes of the 500 simulations; [2] N-Outputs as the sum of the absolute values of all the output fluxes of*
*the 500 simulations; [3] N-stock-changes as the difference between the input and output fluxes of each of the 500*
*simulations; [4] Gaseous emissions are the sum of $N_2O$, $NO$, $N_2$ and $NH_3$ fluxes; [5] Aquatic flux is nitrate leaching*
*($NO_3^-$).*

| | Mean | Std | Median | Q25 | Q75 |
|---|---|---|---|---|---|
| | [kg N ha⁻¹ yr⁻¹] | [kg N ha⁻¹ yr⁻¹] | [kg N ha⁻¹ yr⁻¹] | [kg N ha⁻¹ yr⁻¹] | [kg N ha⁻¹ yr⁻¹] |
| | | | | | |
| N-Inputs[1] | 212.3 | 9.1 | 215.2 | 203.3 | 220.0 |
| N-Outputs[2] | 198.3 | 11.2 | 196.4 | 191.4 | 204.0 |
| N-stock-changes[3] | 13.8 | 2.1 | 13.7 | 14.5 | 12.5 |



| | | | | | |
|---|---|---|---|---|---|
| Input fluxes | | | | | |
| N deposition | 6.3 | 0.8 | 6.8 | 6.0 | 6.8 |
| Bio. N fixation | 45.6 | 4.3 | 45.7 | 43.7 | 47.7 |
| N in min. fertilizer | 80.2 | 4.8 | 81.3 | 76.6 | 82.7 |
| N in organic fertilizer | 80.2 | 3.6 | 80.9 | 77.5 | 82.7 |
| | | | | | |
| Output fluxes | | | | | |
| Gaseous emissions[4] | 55.4 | 8.8 | 55.1 | 48.9 | 61.6 |
| $N_2O$ | 2.6 | 0.8 | 2.5 | 2.1 | 3.1 |
| NO | 3.2 | 1.5 | 2.9 | 2.0 | 4.1 |
| $N_2$ | 15.5 | 7.0 | 14.6 | 9.9 | 20.7 |
| $NH_3$ | 34.0 | 6.7 | 31.8 | 29.3 | 36.9 |
| Aquatic fluxes[5] | | | | | |
| $NO_3$ leaching | 14.1 | 4.5 | 13.6 | 11.0 | 17.0 |
| | | | | | |


The N influx was composed by the input of synthetic fertilizer of 80.2 ± 4.8 kg N ha$^{-1}$ yr$^{-1}$ (IQR
76.6 to 82.7) and organic fertilizer of 80.2 ± 3.6 kg N ha$^{-1}$ yr$^{-1}$ (IQR from 77.5 to 82.7), followed
by the biological nitrogen fixation (BNF) via legumes estimated as 45.6 ± 4.3kg N ha$^{-1}$ yr$^{-1}$ (IQR
from 43.7 to 47.7) and nitrogen deposition of 6.3 ± 0.8kg N ha$^{-1}$ yr$^{-1}$ (IQR from 6.0 to 6.8). Thus,
almost 75% of the nitrogen input influx is related to the fertilization (mineral and organic) whilst
the minor part that corresponds to nitrogen fixation and deposition approximates to 25%.
The N outflux consist of gaseous N fluxes of 55.4 ± 8.8 kg N ha$^{-1}$ yr$^{-1}$ (IQR from 48.9 to 61.6),
aquatic N fluxes (only nitrate leaching into surface waters was considered) of 14.1 ± 4.5 kg N
ha$^{-1}$ yr$^{-1}$ (IQR from 11.0 to 17.0), N fluxes by removed N in yields, straw and feed of 128.8 ±
8.5 kg N ha$^{-1}$ yr$^{-1}$ (IQR of 125.2 to 131.7) (see Figure 4 and Table 5). Based on the
aforementioned results all gaseous and aquatic N-fluxes correspond to about 28% and 7% of
the N output flux respectively, while the far largest N output flux was N removed in yields, straw
and feed representing almost 65% of the N outflux (Figure 4).





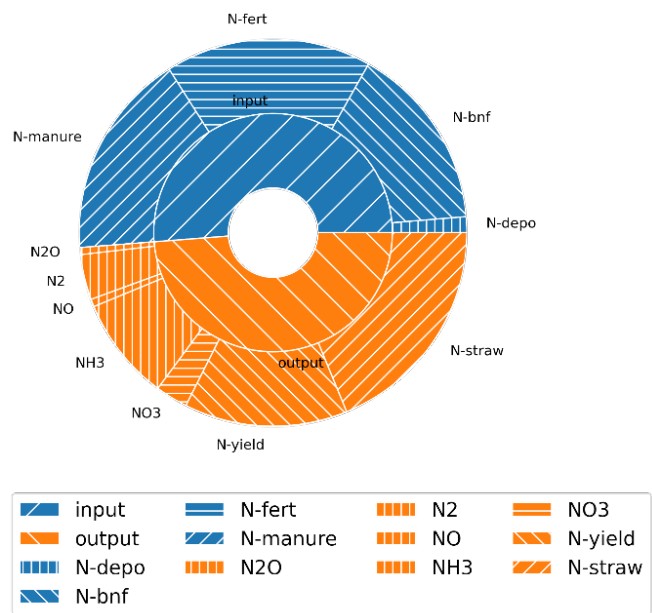


Figure 4. Averaged annual nitrogen balance (inner ring of the pie diagram) and their decomposition into the various
components of the N fluxes (outer ring of the pie diagram); (all data summarized in Table 5).


The simulated gaseous fluxes were composed of $N_2O$ emissions estimated to 2.6 ± 0.8 kg
$N_2O$-N ha$^{-1}$ yr$^{-1}$ (IQR from 2.1 to 3.1), NO emissions of 3.2 ± 1.5 kg NO-N ha$^{-1}$ yr$^{-1}$ (IQR from
2.0 to 4.1), $N_2$ emissions 15.5 ± 7.0 kg $N_2$-N ha$^{-1}$ yr$^{-1}$ (IQR range from 9.9 to 20.7) and $NH_3$
emissions of 34.0 ± 6.7 kg $NH_3$-N ha$^{-1}$ yr$^{-1}$ (IQR from 29.3 to 36.9). Ammonia volatilization
represents the largest share (61.48%) of gaseous N losses, with highest densities in the
emission distribution between approx. 25 and 35 kg N ha$^{-1}$, followed by di-nitrogen losses
(28.03%) of gaseous N losses, with a much wider emission variability in the distribution,
followed by $NO_3$ (5.79%) and $N_2O$ (4.7%). Figure 5 shows the overall NB in a waterfall diagram
adding up cumulative all in- and out-fluxes illustrating the uncertainty distribution of each flux
contributions. The waterfall diagram illustrates the overall outcome of the NB, a N accumulation
into the soil as the difference between all out-fluxes minus all in-fluxes.



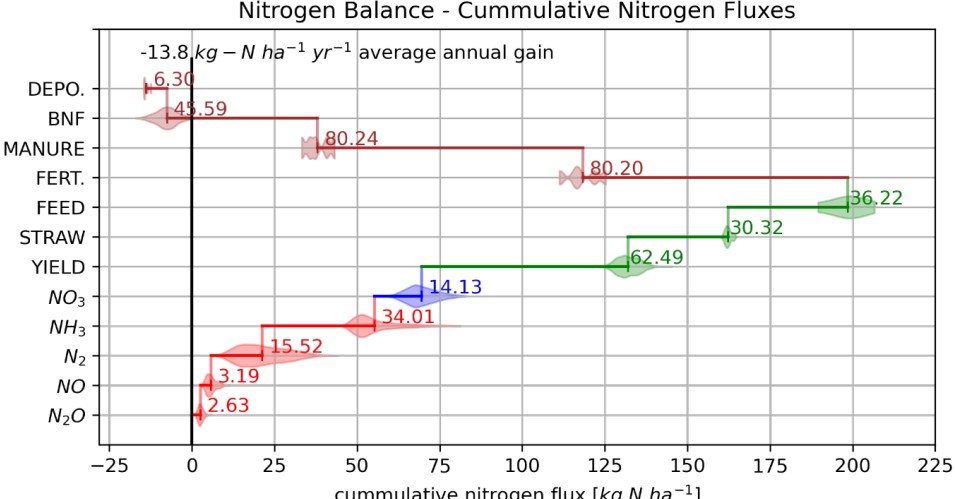


*Figure 5. Waterfall representation of the result distributions of the different Nitrogen in- and outfluxes of the cropland*


*cultivation in Thessaly. Vertical lines in the distributions indicate mean values of the corresponding N-flux. Red*


*colors indicate gaseous outfluxes, blue aquatic fluxes, green biomass yield and feed production outfluxes and brown*


*color indicates N influxes such as synth. N-fertilizer, N-Manure, biological N fixation (BNF) and N deposition. The*


*Resulting N sink of the Nitrogen Balance (based on distribution means) is -13.8 kg N ha$^{-1}$ yr$^{1}$. (Negative value*


*indicates flux into the soil).*



Nitrate leaching mean estimates were 14.1 ± 4.5 kg NO$_3$-N ha$^{-1}$ yr$^{-1}$ (IQR from 11.0 to 17.0)
with a bell-shaped distribution.
Total yield and biomass (straw and feed) N export fluxes were 62.4 ± 4.4 kg N ha$^{-1}$ yr$^{-1}$ with
uncertainty ranges from 59.9 to 65.1 consisting of yield N exports (grains and cotton balls) of
30.3 ± 1.7 kg N ha$^{-1}$ yr$^{-1}$ (IQR from 29.6 to 30.9) and for straw and feed N exports of 36.1 ± 6.0
kg N ha$^{-1}$ yr$^{-1}$ (IQR from 34.9 to 37.6). The result distributions for yield N are well bell shaped,
for feed biomass N very moderate bell shaped and well distributed within the bounds and for
straw N very sharp within a comparable small interval.
Figure 5 illustrates the cumulative nitrogen fluxes composing the NB as a waterfall diagram
considering the mean of each component. The NB results in a net N sink of 13.8 kg N ha$^{-1}$ yr$^{-}$
$^{1}$ (see result distribution in Figure 6) for the region corresponding to an annual carbon
sequestration of approx. 0.5 tons C ha$^{-1}$ yr$^{-1}$ as depicted in Figure 2 b) (see also the annual
dynamics of the topsoil (30 cm) soil organic carbon and nitrogen distributions in Figure 8).

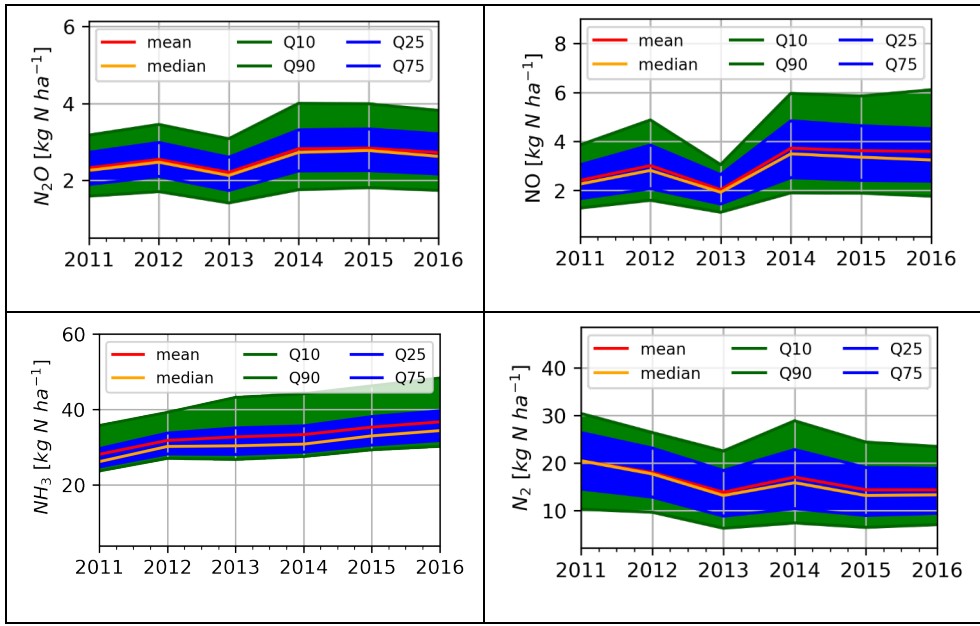

Nitrogen Balance Uncertainty

nitrogen flux $kg\text{-}N\ ha^{-1}\ yr^{-1}$


*Figure 6. Distribution of the overall Nitrogen Balance of the cropland cultivation in Thessaly: Statistical analysis across all 500 individual NB results of the inventory simulations (mean 13.8 kg N ha[-1] yr[-1], median 13.7 kg N ha[-1] yr[-1]) corresponding to the Carbon balance in Figure 2.*


Figure 7 and Figure 8 show the dynamics of the annual distribution of the gaseous and aquatic
outfluxes as well as the dynamics of the annual distributions of the top soil (30 cm) soil organic
carbon and nitrogen pools for the evaluation period 2011 – 2016.

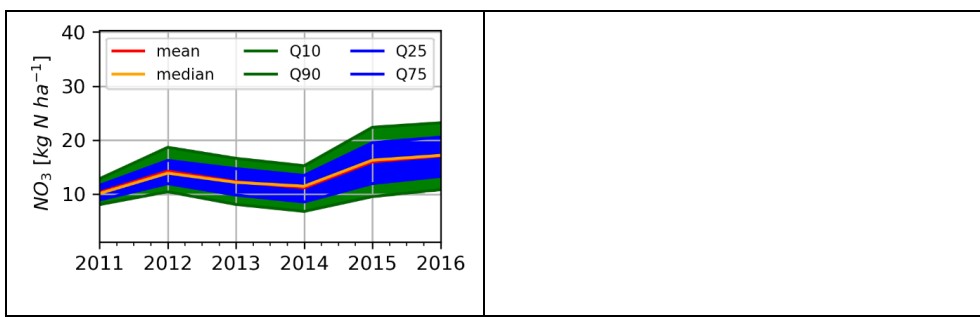

*Figure 7. Annual dynamics of the uncertainty distributions of the gaseous (subfigure a) to d)) and aquatic (subfigure e)) N outfluxes 2011 – 2016. Uncertainty bandwidth (blue band) defined as the range between the q25 and the q75 quartile, green band (Q10. to Q90 interval) indicating the variance of the fluxes neglecting the outliners of the distribution.*

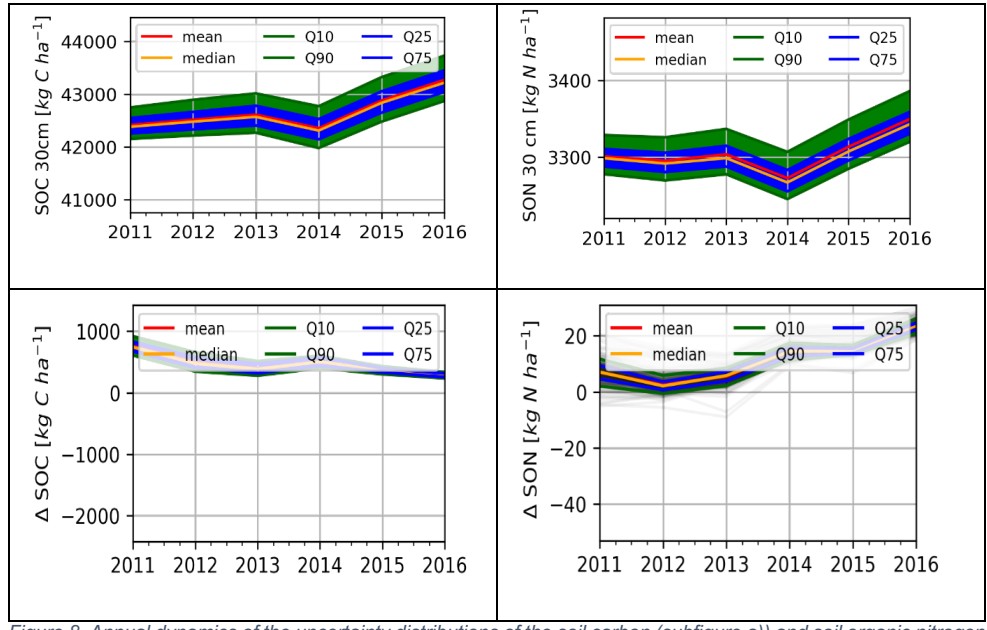

*Figure 8. Annual dynamics of the uncertainty distributions of the soil carbon (subfigure a)) and soil organic nitrogen (subfigure b)) and the corresponding dynamics of the uncertainty distributions of the annual change rates of the total soil carbon and nitrogen pools (subfigures c) and d)) respectively.*

## 4    Discussion.

Simulating the N and C budgets is helpful for the understanding/explanation of the pattern how nutrients are being supplied from the soil to crop as well as the pathways of the excess gaseous and aquatic excess nitrogen fluxes. In this way, improvements on the agricultural practices e.g. N fertilization strategy could be accomplished to sustain agricultural output and



minimize environmental harm. In this study, an assessment of the full C and N balance of a
regional cropland agroecosystem is reported for the first time using inventory simulations with
a process-based ecosystem model in combination with the quantification of the associated
modelling uncertainty. Up to present, process-based modelling studies mainly focus on single
site applications e.g. Daycent: (del Grosso et al., 2005; Gurung et al., 2020), APSIM: (Vogeler
et al., 2013), CERES-EGC: (Dambreville et al., 2008; Gabrielle et al., 2006; Heinen, 2006;
Hénault et al., 2005), CERES-Wheat: (Mavromatis, 2016), DNDC: (Li, 2000),
LandscapeDNDC: (Haas et al., 2013; Klatt et al., 2015a; Molina-Herrera et al., 2016; Zhang et
al., 2015). Fewer studies deploy models on the regional to national (del Grosso et al., 2005;
Kim et al., 2015; Klatt et al., 2015a) or continental to global scale (del Grosso et al., 2009;
Franke et al., 2020; Jägermeyr et al., 2021; Smerald et al., 2022; Thompson et al., 2019). All
of these studies focus in general on one specific or a few components of the carbon or nitrogen
cycle such as e.g. soil carbon stocks or $N_2O$ emissions.
There are very few cases where an attempt for regional estimation of the NB has been made.
Schroeck et al., (2019) reported an assessment of the NB for a large alpine watershed in the
Austrian Alps characterized by arable production in the low-lying areas and grassland in the
mountains. In addition, Lee et al., (2020) tried to estimate nitrogen balances in Switzerland
alternating the cropping systems or management practices. There were, also, cases where
the regional NB was estimated with the use of nitrogen balance equations (He et al., 2018).
In order to achieve a more concrete and complete analysis of the CB and NB that could be
used for future policy development, an uncertainty analysis is considered as
necessary/mandatory. The IPCC guidelines demand for UNFCC reporting the uncertainty
quantification of any reported inventory study (IPCC Updated guidelines 2019). Recent
publications have reported the deployment of different methods to assess and quantify the
various sources of uncertainty in ecosystem modelling. (Klatt et al., 2015b) published a study
on the impact of parameter uncertainty on $N_2O$ emissions and $NO_3$ leaching on the regional
scale. (Houska et al., 2017) deployed the GLUE method (Generalized Likelihood Uncertainty
Estimation) for the LandscapeDNDC model on a grassland site, others studies such as



(Lehuger et al., 2009a; Li et al., 2015; Myrgiotis et al., 2018a) used the Bayesian Model
Calibration and Uncertainty Assessment approach, which has been used in the current study
as well.

4.1 Yield and feed Production
A number of studies including crop yields estimates under different environmental conditions
and crop management options have been published. Molina-Herrera et al., (2016) reported
validation results deploying LandscapeDNDC on cropland and grassland sites across Europe.
The study reported good agreement in reproducing observed above ground biomass and yield
estimates leading to a high trustworthiness of model results. Similar model performance for
the cultivation of commodity crops was reported by (Kasper et al., 2019; Klatt et al., 2015a;
Molina-Herrera et al., 2017; R. J. Petersen et al., 2021)
Voloudakis et al., (2015) simulated cotton production in seven different areas of Greece
applying the AquaCrop model for future climate scenarios. The model was calibrated and the
results were validated with data sets acquired for years 2006 and 2005/2007 respectively from
a site experiment conducted in the area of Karditsa, Thessaly. The observed and simulated
results presented in the study were well matched and in line with the results presented in our
study with averaged yields of 3.65 tons ha$^{-1}$ (mean 3.3 tons ha$^{-1}$ and median 3.5 tons ha$^{-1}$).
Lyra and Loukas, (2021) used REPIC model to estimate the crop growth/yield production of
several crops in the Basin of Almyros, Thessaly. The simulated results were approximately 11
tons ha$^{-1}$ clover, 3.3/3.5 tons ha$^{-1}$ cereals/wheat, 3.8 tons ha$^{-1}$ cotton and 9 tons ha$^{-1}$ maize,
being well compared to the results of the current research shown in Table 3.
The application of AquaCrop in the cotton cultivation of the research of Tsakmakis et al.,
(2019), proved the accurate estimation of the cotton yield when using the default set of
parameters in both cases of growing degree days (GDD) and calendar days (CD) modes for a
site in Northern Greece. For the year of 2015 the harvested seed cotton yield was 3.974 tons
ha$^{-1}$ ± 0.45 and 3.35 tons ha$^{-1}$ ± 0.397 in 2016 with a slight overestimation of 0.018 tons ha$^{-1}$


and 0.026 tons ha⁻¹ while in 2016 there was a marginal underestimation by 0.06 tons ha⁻¹ and
0.046 tons ha⁻¹ for the respective aforementioned cases. The model did not perform well when
the parameter sets were altered based on other studies (García-Vila et al., 2009).
There are few cases in literature concerning yield simulations on a European level. Based on
the yield datasets of FAO and EUROSTAT Ciais et al., (2010a) estimated mean crop yields
for the period 1990–1999 at the scale of EU-25 as 6.1 (FAO) and 5.3 (EUROSTAT) tons DM
ha⁻¹ yr⁻¹, respectively, which corresponds well to results of our study. Haas et al., (2022)
estimated with a model ensemble mean for crop yields for EU-27 of 4.41 ± 1.85 tons DM ha⁻¹
yr⁻¹ for the period 1990–1999. Lugato et al., (2018) estimated cropland yield projections of
4.34 tons DM ha⁻¹ yr⁻¹ (mean), ranging from 3.69 to 4.90 tons DM ha⁻¹ yr⁻¹ with the DayCent
model for EU-27, comparable to the 6.18 tons DM ha⁻¹ yr⁻¹ average simulated crop yields of
this study. The simulated yields in the current study vary from 3.3 to 9.9 tons DM ha⁻¹ yr⁻¹ for
the cases of cotton and maize respectively.
Higher yield estimates for the region of Thessaly in this study are certainly due to the inclusion
of the legume feed crops in the rotations. This argument is supported by a recent meta-analysis
by (Lu, 2020) that concluded that on average yield increases of 5.0 to 25% can be expected
for various conditions if residues are completely returned to the field as compared to no-residue
return systems. Similar results were reported by Fuchs et al., (2020) and Barneze et al., (2020).

4.2   Carbon and Nitrogen Balance:
4.2.1   SOC stocks
Haas et al., (2022) reported results of a European inventory simulation of soil carbon stocks
and $N_2O$ emissions using a model ensemble. The study deployed in a baseline simulation
across EU-27 a similar residues management as compared to our study resulting in very stable
carbon stock dynamics over a long period (1950-2100). In this study, the estimated carbon
sequestration of 0.5 (UA mean and median) ± 0.3 tons C ha⁻¹ yr⁻¹ is mainly caused by the
inclusion of legume feed crops within the crop rotation leading to increased litter production



and C input into the soil (Barneze et al., 2020; Fuchs et al., 2020; K. Petersen et al., 2021).
Haas et al., (2022) reported a management scenario with 100% of crop litter remaining on the
field leading to averaged C-sequestration rates of over 1 ton C ha$^{-1}$ yr$^{-1}$ across EU-27. As the
residues management in this study is between the baseline and buried scenario of Haas et al.,
(2022), our results compare well to results reported in this study.
Other studies such as (Lugato et al., 2014) reported C sequestration rates for the conversion
of cropland into grassland ranging between 0.4 and 0.8 tons C ha$^{-1}$ yr$^{-1}$. Lugato et al., (2014)
reported averaged SOC change rates for a cereal straw incorporation scenario for EU-27 of
0.1 tons C ha$^{-1}$ yr$^{-1}$ (estimates from 2000 to 2020).
The Mediterranean agroecosystems show a winter/summer rainfall and soil cover variation,
spatial diversity, and the longest continuous settlement and dense cultivation by man (Yaalon,
1997). Based on the Greek National Map of SOC (2020) by Triantakonstantis and Detsikas,
(2021), SOC values for the region of Thessaly vary from 22.95 to 86.97 tons ha$^{-1}$ with the lower
values in the main plain of the region and higher values in the croplands closer to the
mountainous areas. Comparing these maps with SOC map of the 30cm topsoil of our story,
we clearly see similar patterns, which relate to the similarity of SOC data used to initialize the
region in LandscapeDNDC.

4.2.2   N$_2$O emissions
This study reported estimates of N$_2$O emissions of 2.6 ± 0.8 kg N$_2$O-N ha$^{-1}$ yr$^{-1}$ (IQR from 2.1
to 3.1) for a mixed crop / legume feed crop rotation. The LandscapeDNDC validation study of
Molina-Herrera et al., (2016) reported for the Italian site Borgo Cioffi (Mediterranean climate,
Ranucci et al., (2011) annual N$_2$O emissions of 2.49 kg N$_2$O-N ha$^{-1}$ yr$^{-1}$ while two sites in
southern France showed annual N$_2$O emissions from 0.52 to 3.34 kg N$_2$O-N ha$^{-1}$ yr$^{-1}$. N$_2$O
emission estimates of our study were higher than results reported by Haas et al., (2022) using
a multi model ensemble estimating average soil N$_2$O emissions from European (EU-27)





cropping systems for the period 1980–1999 of 1.46 ± 1.30 kg $N_2$O-N ha$^{-1}$ yr$^{-1}$ under
conventional (*Baseline*) management and comparable average N input.
Leip et al., (2011) applied the DNDC–Europe model across EU-25 reporting averaged $N_2$O
emissions of 1.8 kg $N_2$O-N ha$^{-1}$ yr$^{-1}$, while on basis of UNFCC reporting and a fuzzy logic
model Ciais et al., (2010b) estimated average $N_2$O emissions from EU-27 croplands at 1.46 ±
0.22 kg $N_2$O-N ha$^{-1}$ yr$^{-1}$.
Klatt et al., (2015a) reported for an inventory (Saxony, Germany) mean $N_2$O emission of 1.43
± 1.25 kg $N_2$O-N ha$^{-1}$ yr$^{-1}$ using a very similar uncertainty quantification approach.
As discussed by Haas et al., (2022) and Janz et al., (2022), increased $N_2$O emissions can be
seen after the addition of crop residues and may be attributed to a stimulation of denitrification
activity by the added substrate and the creation of anaerobic microsites by increased soil
respiration.
Arable land cultivation in Thessaly does not experience strong winter frost and severe soil
freezing such that $N_2$O freeze-thaw emissions as reported by Wagner-Riddle et al., (2017) or
del Grosso et al., (2022) do not play any role in the N budget.
The $N_2$O estimations related to livestock presented in the study of Sidiropoulos and
Tsilingiridis, (2009) varied in a range of 0.74 to 4.33 kt $N_2$O, with an average of 2.84 kt $N_2$O
depended on the average values of emission factors used (for the year 2005). The estimates
were based on the emission factors of IPCC guidelines and the number of animal heads were
derived from the data sets acquired from the Hellenic Statistical Authorities as in the current
study.
Cayuela et al., (2017) conducted a meta-analysis of the direct $N_2$O emissions for a number of
cropping systems for the Mediterranean climate where the emission factors (EFs) were altered
under different fertilization and irrigation conditions. Higher fertilization rates led to higher EFs
(0.82% less than the 1% of IPCC). Additionally, irrigated and intensively cultivated crops had
higher EFs than rainfed (up to 0.91% dependent on the irrigation method). The relatively high
EF of maize in this study could be possibly attributed to the irrigation without the application of
water-saving methods and the on average higher N application rates (Cayuela et al., 2017).




### 4.2.3 Nitrate leaching

This study reported average $NO_3$ leaching fluxes (only nitrate leaching into surface waters) of
14.1 ± 4.5 kg $NO_3$-N ha$^{-1}$ yr$^{-1}$. Molina-Herrera et al., (2016) reported for the LandscapeDNDC
validation study cropland nitrate leaching fluxes of approx. 7 to 88 kg $NO_3$-N ha$^{-1}$ yr$^{-1}$. In
addition, in the research of Molina-Herrera et al., (2017) the described $NO_3$ leaching results
varied from 13 to 8 kg $NO_3$-N ha$^{-1}$ yr$^{-1}$ from 2009 to 2012 showing higher values in regards to
the precipitation and fertigation. The most comparable site Borgo Cioffi resulted in a
comparable annual $NO_3$ leaching flux of 18.62 kg $NO_3$-N ha$^{-1}$ yr$^{-1}$. de Vries et al., (2011)
estimated the $NO_3$ leaching with the use of four different models with varying values from 5 to
40 kg $NO_3$-N ha$^{-1}$ yr$^{-1}$ for the area of our study. These high values could be explained by the
fact that it corresponds both to groundwater and runoff. Klatt et al., (2015b) reported in an
uncertainty assessment for a regional inventory (Saxony, Germany) leaching rates of 29.32 ±
9.97 kg $NO_3$-N ha$^{-1}$ yr$^{-1}$ for a wheat-barley-rapeseed rotation simulated by the LandscapeDNDC
model. The agricultural system and management regime is comparable; higher $NO_3$ leaching
rates were most likely due to higher N fertilization rates (up to 150 kg N) compared with higher
annual precipitation in the region leading to more intense percolation and therefore to stronger
leaching of available $NO_3$ while in our study the fertilization regime was more lean (up to
average of 80 kg N input) such that soil nutrient competition was higher and available nitrate
was more likely to be immobilized by plant uptake.

### 4.2.4 NO emissions

In the current study, the model estimated NO emissions were in average 3.2 ± 1.5 kg NO-N
ha$^{-1}$ yr$^{-1}$. Butterbach-Bahl et al., (2009) performed the very first European inventory of soil NO
emissions using a modified version of DNDC reporting low NO emission rates mostly below 2
kg NO-N ha$^{-1}$ yr$^{-1}$. Molina-Herrera et al., (2017) recently reported a full NO emission inventory
for the State of Saxony Germany compiling annual NO emissions from agricultural soils



ranging from 0.19 to 6.7 kg NO-N ha$^{-1}$ yr$^{-1}$ simulated by LandscapeDNDC. The study reported
the model performance on simulating soil NO emissions on more than 20 different sites. The
study of Schroeck et al., (2019) reported for a regional inventory of arable soils in Austria
simulated by LandscapeDNDC annual NO emissions of 1.0–1.5 kg NO-N ha$^{-1}$ (for the year
2000), while empirical approaches such as Stehfest and Bouwman, (2006) estimated emission
of similar magnitude. Zhang et al., (2015) reported in a model inter-comparison and validation
study of NO and $N_2O$ fluxes including three ecosystem models, consistent simulation results
for the LandscapeDNDC model with NO emission strengths of cropland soils were between 1
and 3 kg NO-N ha$^{-1}$ yr$^{-1}$ across the sites.

4.2.5   $NH_3$ emissions
Schroeck et al., (2019) stated that validation studies of $NH_3$ volatilization for any
biogeochemical model were very rarely reported in literature, mainly due to the complexity and
a lack of flux observations at spatial and temporal high resolution.
In our study the assessment of soil $NH_3$ emissions of 34.0 ± 6.7 kg $NH_3$-N ha$^{-1}$ yr$^{-1}$. High $NH_3$
volatilization and emission rates can be explained by the predominating neutral to basal soils
conditions (pH values of 7 and above) in the study region favouring the Henry $NH_4$/$NH_3$
equilibrium towards higher $NH_3$ gases enabling ammonia to diffuse out of the soil into the free
atmosphere.
The IPCC emission factor (EF) method for $NH_3$ volatilization reports estimates of 20% of N
input into the soil to be volatilized as $NH_3$. For our study, IPCC methodology for $NH_3$ would
lead to 16 kg $NH_3$-N ha$^{-1}$ yr$^{-1}$, which is approximately half the emission strength of the
simulated result. This is due to the neglection of soil properties in the IPCC EF approach which
in contrast is reflected in the modelling approach.
Sidiropoulos and Tsilingiridis, (2009) estimated a national livestock originated $NH_3$ value of
about 40 kt $NH_3$ yr$^{-1}$ of the total 73 kt $NH_3$ yr$^{-1}$ for Greece and the year 2005, which corresponds



to approx. 22 kg ha$^{-1}$ yr$^{-1}$ for the region of Thessaly (the arable land in the region accounts for
20% of the national).
Ramanantenasoa et al., (2018) compared two methods CADASTRE_NH3 and the one of
CITEPA in France on a regional and national level in order to estimate the NH$_3$ emissions and
emission factors. CADASTRE_NH3 is a combination of spatial and temporal databases of
meteorological, soil and N fertilizing data with the process-based Volt'Air model, on small
agricultural regions scale. CITEPA is the organism responsible for the national inventories
where the applied methodology is using a number of statistical data excluding the synthetic
and organic fertilizer properties as well as the cultural practices. Their first model gave lower
results than the second by 29%, being, though, higher than the reported in literature range of
uncertainties. This difference was mainly explained as a difference in the observed applied N
and ammoniacal-N (TAN) giving lower estimates of CADASTRE_NH3 emissions by 63% for
the applications of the organic manure.
There is a number of national NH$_3$ inventories which could be considered detailed and well-
studied like the ones in Denmark, Netherlands, Europe, UK and US. In Denmark, (Geels et al.,
2012) used the DAMOS model to estimate the Danish NH$_3$ emissions (crop, grass and manure
manipulation) where the values ranged in the 5 regions under study from a very small quantity
to 17.4 kg NH$_3$-N ha$^{-1}$ yr$^{-1}$. In the case of the Netherlands (Velthof et al., 2012), a method was
applied based on the estimation of total N excretion and the Total Ammoniacal N (TAN)
percentage in the later. The total national estimated NH$_3$ emission was 88.8 Gg NH3–N for the
year 2009, of which the majority (87%) was related to the housing and manure estimation. de
Vries et al., (2011) used four N budget models of varying complexity for the estimation of the
most common N fluxes in EU27. In the case of NH$_3$ emissions all the models give very
comparable results, which based on their spatial distribution varied from 0-10 kg NH$_3$-N ha$^{-1}$
yr$^{-1}$ for our region under study. The result might differ from our estimated simulation since it is
based on a set of emission factors.
As discussed by Sutton et al., (2013) the majority of the NH$_3$ emissions come as a result of the
agricultural production and are considerably impacted by climate influence. In the case of NH$_3$



volatilization, it could almost double every 5°C temperature given certain complex
thermodynamics dissociation and solubility, whilst soil $NH_3$ emission is influenced by the
available water quantity allowing the $NH_x$ dissolution and use by microbial organisms, which is
afterwards leading to decomposition.

### 4.2.6 Full N balance
At present, the studies of Schroeck et al., (2019) and Lee et al., (2020) are the only to be found
by Web of Science under the search key words "nitrogen AND balance AND process AND
based AND modelling" reporting a compilation of the nitrogen balance for a site or region
applying a process-based ecosystem model even though IPPC is explicitly demanding such
attempts.
Leip et al., (2011) reported the first nitrogen balance for Europe following mixed approach
combining the CAPRI (Common Agricultural Policy Regionalised Impact) model (a global
economic model for agriculture) with different approaches estimating various nitrogen fluxes
in arable land cultivation. The approach e.g. lacks to explicit quantification of the gaseous N
fluxes. The study of Schroeck et al., (2019) overcame this hurdle and applied the process-
based ecosystem model LandscapeDNDC to estimate the full regional nitrogen budgets of
different ecosystems (cropland, grassland and pastures) and climatic zones of a water shed in
Austria. That is a considerable contribution to the attempt for estimating all the N fluxes
possible since only a few countries could offer measurement networks, which could supply
inventory estimates for independent validation Ogle and Paustian, (2005).
The $N_2O$ estimate in Schroeck et al., (2019) and the current study is of a comparable level. In
the later research, the estimated value was 2.6 kg N ha$^{-1}$ yr$^{-1}$ while Schroeck et al., (2019)
reports 1.51 kg N ha$^{-1}$ yr$^{-1}$ lower about 40%. The NO fluxes differ by far since we reported a
mean value of 3.2 kg NO-N ha$^{-1}$ yr$^{-1}$ while Schroeck et al., (2019) reports 0.08 kg NO-N ha$^{-1}$
yr$^{-1}$. This is related to some recent model advances, which have been made during this study,
which elevated the NO production in LandscapeDNDC (Molina-Herrera et al., 2017). Ammonia



volatilization differs, also, substantially between the two studies. Our study reports 34 kg $NH_3$-
N $ha^{-1}$ $yr^{-1}$ and Schroeck et al., (2019) moderate emissions of 0.23 kg $NH_3$-N $ha^{-1}$ $yr^{-1}$. The
stronger $NH_3$ volatilization in our study is mostly driven by the high pH-values of the soils in
the region of Thessaly (pH values from 6.5 to 8.2 with a considerable spatial variation, Greek
Soil Map, 2015) and the comparable high manure inputs of arable system in our study, while
in the research of Schroeck et al., (2019) the manure was preferably applied to the grassland
systems and mineral fertilizers to the arable land. Concerning the $NO_3$, Schroeck reported 45.3
kg $NO_3$-N $ha^{-1}$ $yr^{-1}$ which 3 times higher compared to this study (14.1 kg N $ha^{-1}$ $yr^{-1}$)
considering the N- input of approximately 140 kg and 212.3 kg N $ha^{-1}$ $yr^{-1}$ respectively.
The N balance modelling study of Lee et al., (2020) is estimating for Switzerland a national
cropland N balance using an upscaling method based on process-based site simulations with
the DayCent model differentiating the management of the considered cropping systems e.g.
fertilizer rates, tillage or land cover change. The study reported for conventional cultivations
(averaged across 20 years) yield related N outputs accounting for about 60%, NO3- leaching
36.1% and gaseous N emissions 4.1% of the total N outputs. Although the yield related N
output is in accordance with our result of 64.95% there seems to be a discrepancy in the
gaseous and aquatic N fluxes contribution, 27.94% and 7.11% respectively, which could
possibly occur due to the differences in the soil and climatic conditions, or the arable crops
included in the respective researches.
Velthof et al., (2009) used the MITTERA-EUROPE model/method, based on the concoction of
GAINS and CAPRI models, to estimate N fluxes of European soils on NUTS2 scale with the
use of European datasets and literature coefficients, where the fertilizer application and
management was similar to our methodology. The average N Input-Output balance was
calculated as 117 kg N $ha^{-1}$ $yr^{-1}$ composed by manure of 49 kg N $ha^{-1}$ $yr^{-1}$, synthetic fertilizer of
58 kg N $ha^{-1}$ $yr^{-1}$ (in the current study for both cases 80.2 kg N $ha^{-1}$ $yr^{-1}$), biological nitrogen
fixation of 2 kg N $ha^{-1}$ $yr^{-1}$ (our research 45.6 kg N $ha^{-1}$ $yr^{-1}$) and N deposition of 7 kg N $ha^{-1}$
(current study 6.3 kg N $ha^{-1}$ $yr^{-1}$). In contrast to our study the reported output fluxes for $NH_3$ of
8 kg $NH_3$-N $ha^{-1}$ $yr^{-1}$, $N_2O$ of 2 kg $N_2O$-N $ha^{-1}$ $yr^{-1}$, $NO_x$ of 2 kg $NO_x$-N $ha^{-1}$ $yr^{-1}$, $N_2$ of 51 kg $N_2$-N
$ha^{-1} yr^{-1}$ and $NO_3$ leaching of 7 kg $NO_3$-N $ha^{-1} yr^{-1}$ while the differences with the results presented
in our study are $NH_3$ of 34.0 kg $NH_3$-N $ha^{-1} yr^{-1}$, $N_2O$ of 2.6 kg $N_2O$ -N $ha^{-1} yr^{-1}$, $NO_x$ of 3.2 kg
$NO_x$ -N $ha^{-1} yr^{-1}$, $N_2$ of 15.5 kg N $ha^{-1} yr^{-1}$ and $NO_3$ leaching of 14.1 kg $NO_3$-N $ha^{-1} yr^{-1}$.
Additionally, the yield output is estimated as 48 kg N $ha^{-1} yr^{-1}$. The difference with the results
presented in our study, could be related to the different input data used, based on regional
statistics and the use of a biogeochemical model and not on literature factors as in the later
study.
He et al., (2018) assessed the soil N balance for a time spam between 1984 to 2014 based on
the N budget equations (N input – N output) using multiple coefficients from literature in order
to estimate the nitrogen input and output fluxes of six grouped regions in China. The used
datasets were acquired from national Authorities and include cropping land and yields,
synthetic fertilizers, animal heads, soil types etc. The N synthetic fertilizer input is in average
182.4 kg N $ha^{-1}$ and the organic fertilizer of 97.3 kg N $ha^{-1}$, N fixation is estimated as 16.8 kg
N $ha^{-1}$ and the atmospheric deposition as 22 kg N $ha^{-1}$. Almost half of the total averaged N
output losses, 48.9%, was attributed to crop uptake while the respective gaseous losses were
$N_2$ 19.9%, volatilized $NH_3$ 17.3%, $N_2O$ 1.2% and NO 0.7%. As for the $NO_3$ leaching share was
5.8% of the total output N fluxes. The previous results are comparable to the results of the
current study mainly in the aquatic fluxes, which account for approx. 7%, as described in Figure
4. The difference that appears in the N uptakes could be a result of the fact that in our study it
includes the yield, straws and feed.
Myrgiotis et al., (2019) reported a $N_2O$ emission factor (EF) estimate for arable land of 0.59%
and associated uncertainty bands of ± 0.36% which is half of the $N_2O$ EF of our study of approx.
1.2% (data not shown). The reported $NO_3$ leaching factor (LF) mean for their region was 14%
(±7 %). Myrgiotis reported an averaged NUF of 37 % (±7 %) which is almost half of the NUF
of 67.3% we reported in the current study.
As reported in OECD (OECD Nutrient Balance, 2020) the averaged nitrogen input rate for
Greece is estimated at about 290 kg N $ha^{-1}$ for the years 2010-2015. Based on the regional
land share (~20% of the national arable land) the nitrogen balance of the area under study





becomes 11.6 kg N ha$^{-1}$ yr$^{-1}$. As presented in Table 5 the simulated mean nitrogen balance
results in an in-flux of 13.8 kg N ha$^{-1}$ yr$^{-1}$ (IQR 11.9 to 16.0) which is well in line with the
aforementioned OECD value.

4.3   Uncertainty Analysis and Quantification
The MCMC algorithm was used by Santabarbara (2019) to estimate the joint parameter
distribution of the fundamental bio-geochemical process parameters in LandscapeDNDC
when simulation soil C and N fluxes. Propagating these joint parameter distributions through
the model (by sampling 500 joint parameter distributions and performing inventory simulations
with each parameter set with the model) for estimating the regional C and N fluxes was leading
to distributions for any model result on the regional scale. Statistical analysis calculating mean,
median as well as the interquartile range (Q25 to Q75) determines best estimates and the
uncertainty range of any model output on the regional scale, demonstrating the superiority of
the method for assessing any ecosystem response by modelling instead of reporting single
results. This is a novel approach, that to our knowledge has not been reported before in
literature for the carbon and nitrogen balance and neither been applied to regional simulations
by any process-based model.
In this study, the estimated UA mean and median of the carbon sequestration of $0.5 \pm 0.3$ tons
C ha$^{-1}$ yr$^{-1}$ is associated with an uncertainty range from 0.4 to 0.7 tons C ha$^{-1}$ yr$^{-1}$ which
compares well to the spatial uncertainty of C-sequestration in the study of Haas et al., (2022).
The approach used in this study enabled to assess the carbon and nitrogen balance of the
cropland ecosystems including an assessment of the prediction uncertainty.
van Oijen et al., (2005) used the Bayesian calibration method to acquire the parameter
posterior probability distribution to sample from, for simulation of the model results. Lehuger et
al., (2009b) used the Bayesian calibration method for the enhancement of the CERES-EGC
model parameterization (reduction of the apriori parameter distribution) as well as
quantification of the uncertainty of the simulated N$_2$O emissions in different sites. The



estimated fluxes of the different sites resulted in a range between 0.088 to 3.672 kg $N_2O$-N ha$^{-1}$ yr$^{-1}$ with values for the q05 quantile of 0.066 to 0.115 kg $N_2O$-N ha$^{-1}$ yr$^{-1}$ and for the Q95

quantile from 1.676 to 5.874 kg $N_2O$-N ha$^{-1}$ yr$^{-1}$ with an averaged value of 1.04 kg $N_2O$-N ha$^{-1}$
yr$^{-1}$ which is lower than the result of the current study but still in the same order of magnitude.
Klatt et al., (2015b) quantified a parameter-induced uncertainty analysis on the regional scale
applying the LandscapeDNDC model for simulating $N_2O$ emission and $NO_3$ leaching
inventories similar to our study. The region was represented by 4000 polygons of arable land
(state of Saxony, Germany) for crop rotations of barley, wheat and rapeseed. The investigated
model parameter related uncertainties give a high confidence for the parameter use in our
research. The results of Klatt et al., (2015b) display a likelihood range of 50% (the IQR range
between Q25 and Q75) for $N_2O$ emissions from 0.46 to 2.05 kg $N_2O$-N ha$^{-1}$ yr$^{-1}$ which is in
good comparison to our results of 2.1 to 3.1 kg $N_2O$-N ha$^{-1}$ yr$^{-1}$. The average direct $N_2O$
emissions are 1.43 kg $N_2O$-N ha$^{-1}$ yr$^{-1}$ comparable to the result of our study (mean: 2.6 and
median: 2.5 kg $N_2O$-N ha$^{-1}$ yr$^{-1}$ across approx. 1000 polygons). As for leached $NO_3$, Klatt et
al., (2015b) reported leaching rates of mean value: 29 kg $NO_3$-N ha$^{-1}$ yr$^{-1}$, (IQR from 24.5 to
36.0), which is higher compared to the results of our study: Mean: 14.1 kg $NO_3$-N ha$^{-1}$ yr$^{-1}$,
median: 13.6 kg $NO_3$-N ha$^{-1}$ yr$^{-1}$ (IQR from 11 to 17).
Butterbach-Bahl et al., (2022) reported the influence of management uncertainties for
compiling national inventories of $CH_4$ and $N_2O$ emission from various rice cultivation systems
in Vietnam. The study applied a sampling technique varying model input data within a given
range and analyzing the influence on the assessed $CH_4$ and $N_2O$ emission strengths. As the
underlying cropland systems were fundamentally different, the assessed uncertainty ranges
were comparable and the study is supporting our approach to focus on reporting uncertainty
ranges rather than single values.
The current study postulates a novel approach to report regional scale C and N fluxes from
process-based models to become the standard reporting method fulfilling the long-time
demanded reporting requirements of the IPCC (IPCC Guidelines and IPCC 2019 updates on
Guidelines). Additionally, instead of focusing only on topics limited to soil carbon stocks





dynamics or greenhouse gas emissions, we propose the report of the full C and N balance
including all components of the various fluxes e.g. gaseous and/or aquatic being available by
the individual process-based models. This constitutes an effective method to assess the
environmental impact of the crop production on a national scale, therefore optimize the farming
practices, and suggest possible solution for sustainable agriculture development.


5     Conclusion

In this research, we presented for the first time a regional inventory of the full carbon and
nitrogen balance including all sub-components of these fluxes simulated by a process-based
model. Additionally, the study has fulfilled the demand to report always the associated
uncertainties for any modelling results being published in literature. This supports the
trustworthiness of the reported results. The LandscapeDNDC model is applied in the region of
Thessaly after using the MCMC Bayesian calibration method against soil, daily climatic and
crop management regional datasets. The main scope/goal was the assessment of the total C
and N balance that enhance the efforts towards the understanding of the cropland system and
the respective interactions within the C and N balances.
Observed GHG emission datasets are scarce if not unavailable. Thus, the modelled yield
results were evaluated/validated against the observed values of crop yields provided by the
Hellenic Statistical Authority and showed a good fit for almost all simulated crops except for
the case of maize where there was a slight underestimation.
In addition, a full uncertainty analysis is presented based on the Metropolis-Hastings algorithm
where a parameter subset and input data preturbation was sampled and simulated in 500
iterations resulting in a final probability density function (PDF) for each one of the N and C
balance fluxes building a full uncertainty analysis of the modelled results. This helps to build
trustworthiness in modelling assessments and estimates.





All of the above constitute the novelty of the conducted research that could be further
elaborated by a number of proposed mitigation measures, which could help in the abatement
of the GHG emissions and N fluxes from crop/agricultural land.

6    Aknowledgements
The author Odysseas Sifounakis received a Ph.D. research scholarship from Alexandros S.
Onassis Public Benefit Foundation, Greece, part of which is the research presented in the
current publication.

7    Code/Data availability
The LandscapeDNDC model source code is available via Butterbach-Bahl, Klaus; Grote,
Rüdiger; Haas, Edwin; et al. (2021): LandscapeDNDC (v1.30.4). Karlsruhe Institute of
Technology (KIT). DOI: 10.35097/438
All publication results (tables and data for figures) will be made available in the supplementary
material associated with this paper.


8    Author contributions
Mr. Odysseas Sifounakis has conceived and designed the analysis and collected the data. He,
also, performed the analysis and wrote the paper.
Dr. Edwin Haas conducted research and wrote the paper.
Prof. Dr. Klaus Butterbach-Bahl substantially contributed to research planning, manuscript
writing and editing and, also, provided funding opportunities.
Prof. Dr. Maria P. Papadopoulou substantially contributed to research planning, manuscript
writing and editing, and provided funding opportunities.



## 9 Competing interests

All authors have reviewed and accepted the submitted version and declare no conflicts of interest related to this publication.

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





**11  Appendix**
**11.1  Material and Methods**
**Sensitivity Index**
In the first step, the Sensitivity Index algorithm (SI) (Pannell, 1997) was calculated for all
process parameters by splitting the parameter ranges into 10 equidistant values from minimum
to maximum and by rating SI values:
$$SI = \frac{CUM_{max} - CUM_{min}}{CUM_{max}}$$

where $CUM_{max}$ and $CUM_{min}$ are the maximum and minimum cumulative results of 10
simulations. High SI values explain a high sensitivity of the underlying parameter with respect
to the model results, whereas low values or even zero indicates low or no sensitivity.

**11.2  Results**
*Table A 1. Observed yield rates in the region of Thessaly. Cotton yields are the cotton bolls, clover feed is the total*
*harvested above ground biomass, for wheat and barley it is the grain yield, maize is accounted grain ear and the*
*stems Source ELSTAT.*

| Crop Yields [tons dry matter ha⁻¹] | | | | | | |
|---|---|---|---|---|---|---|
| Crops | 2012 | 2013 | 2014 | 2015 | 2016 | Mean |
| Cotton | 2.7 | 3.6 | 3.5 | 3.4 | 3.3 | 3.3 |
| Clover | 8.6 | 8.9 | 8.7 | 7.9 | 7.7 | 8.4 |
| Wheat | 3.3 | 3.3 | 3.3 | 3.7 | 3.6 | 3.4 |
| Barley | 3.2 | 3.2 | 3.2 | 3.5 | 3.5 | 3.3 |
| Maize | 10.9 | 12.1 | 12.3 | 12.7 | 12.1 | 12.0 |


*Table A 2. Crop rotation scenarios (R1 – R5) for the region of Thessaly where the crop abbreviations corn, wiwh,*
*perg, cott and wbar refer to maize, winter wheat, clover (legume feed crops s.a. alfalfa or vetch), cotton and winter*
*barley respectively.*

| years | R1 | R2 | R3 | R4 | R5 |
|---|---|---|---|---|---|
| 2010 | corn | wiwh | perg | cott | wbar |
| 2011 | wiwh | perg | cott | wbar | corn |
| 2012 | perg | cott | wbar | corn | wiwh |
| 2013 | cott | wbar | corn | wiwh | perg |





| 2014 | wbar | corn | wiwh | perg | cott |
| 2015 | corn | wiwh | perg | cott | wbar |
| 2016 | wiwh | perg | cott | wbar | corn |


*Table A 3. Carbon Balance (totals) Summary of the Assessment and Uncertainty Analysis of the of cropland*
*cultivation of the region of Thessaly, Greece, GPP gross primary productivity, TER terrestrial ecosystem respiration,*
*Biomass export includes all C in yield, straw and feed exported from the fields, 360000 ha cropland.*

|  | Mean | Std | Median | Q25 | Q75 |
|---|---|---|---|---|---|
|  | [mio. tons C yr⁻¹] | [mio. tons C yr⁻¹] | [mio. tons C yr⁻¹] | [mio. tons C yr⁻¹] | [mio. tons C yr⁻¹] |
|  |  |  |  |  |  |
| C-Inputs | 4.51 | 0.20 | 4.45 | 4.36 | 4.69 |
| C-Outputs | 4.32 | 0.17 | 4.31 | 4.19 | 4.45 |
| SOC-changes | 0.19 | 0.11 | 0.20 | 0.14 | 0.27 |
|  |  |  |  |  |  |
| Input fluxes |  |  |  |  |  |
| GPP | 4.25 | 0.20 | 4.21 | 4.11 | 4.42 |
| C in manure | 0.25 | 0.01 | 0.26 | 0.25 | 0.26 |
|  |  |  |  |  |  |
| Output fluxes |  |  |  |  |  |
| TER | 3.08 | 0.16 | 3.06 | 2.97 | 3.20 |
| Biomass export | 1.24 | 0.05 | 1.24 | 1.21 | 1.27 |
|  |  |  |  |  |  |


*Table A 4 Nitrogen balance (totals) Summary of the Assessment and Uncertainty Analysis of the total Nitrogen*
*Balance of cropland cultivation of the region of Thessaly, Greece.*

|  | Mean | Std | Median | Q25 | Q75 |
|---|---|---|---|---|---|
|  | [kt-N yr⁻¹] | [kt-N yr⁻¹] | [kt-N yr⁻¹] | [kt-N yr⁻¹] | [kt-N yr⁻¹] |
|  |  |  |  |  |  |
| N-Inputs | 76.5 | 3.2 | 77.8 | 73.3 | 79.1 |
| N-Outputs | 71.7 | 3.2 | 71.2 | 69.4 | 73.7 |
| N-stock-changes | 4.8 | 0.0 | 6.6 | 3.9 | 5.4 |
|  |  |  |  |  |  |
| Input fluxes |  |  |  |  |  |
| N deposition | 2.0 | 0.3 | 2.1 | 1.9 | 2.1 |
| Bio. N fixation | 16.7 | 1.6 | 16.7 | 15.9 | 17.5 |
| N in min. fertilizer | 28.9 | 1.7 | 29.3 | 27.6 | 29.8 |
| N in organic fertilizer | 28.9 | 1.3 | 29.2 | 27.9 | 29.8 |


| | | | | | |
|---|---|---|---|---|---|
| Output fluxes | | | | | |
| Gaseous emissions[1] | 21.2 | 3.1 | 21.1 | 18.9 | 23.4 |
| N$_2$O | 0.9 | 0.3 | 0.9 | 0.7 | 1.1 |
| NO | 1.1 | 0.5 | 1.0 | 0.7 | 1.4 |
| N$_2$ | 4.9 | 2.4 | 4.5 | 2.9 | 6.6 |
| NH$_3$ | 14.3 | 2.6 | 13.5 | 12.5 | 15.6 |
| Aquatic fluxes[2] | | | | | |
| NO$_3$ leaching | 3.9 | 1.3 | 3.8 | 3.0 | 4.7 |
| | | | | | |

1) Gaseous emissions are the sum of N2O, NO, N2 and NH3 fluxes; 2) Aquatic flux is nitrate leaching (NO3-)

*Table A 5. Total crop yields per cultivar and year.*

| Crop Yields [tons dry matter] | | | | | |
|---|---|---|---|---|---|
| Crops | 2012 | 2013 | 2014 | 2015 | 2016 | Mean |
| Cotton | 303 676.9 | 374 424.6 | 359 806.7 | 322 292.0 | 285 780.3 | 329 196.1 |
| Clover | 302 753.2 | 319 401.7 | 338 134.6 | 341 938.4 | 360 693.9 | 332 584.4 |
| Wheat | 477 700.7 | 461 875.5 | 395 902.1 | 430 014.4 | 450 254.3 | 443 149.4 |
| Barley | 84 520.8 | 99 091.8 | 139 402.9 | 139 990.8 | 102 454.7 | 113 092.2 |
| Maize | 332 531.6 | 431 324.6 | 377 783.9 | 351 285.4 | 334 277.7 | 365 440.6 |

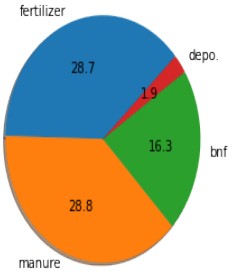

*Figure 9. Shares of components of the annual nitrogen in- and output fluxes.*

*Table A 6. Simulated crop yields per cultivar and year for the irrigated land.*

| Crop Yields [tons dry matter ha$^{-1}$] |
|---|





| Crops | Median | Mean | STD |
|---|---|---|---|
| Cotton | 4.0 | 3.7 | 0.9 |
| Clover | 9.8 | 9.6 | 0.6 |
| Wheat | 3.9 | 3.6 | 0.9 |
| Barley | 5.3 | 5.0 | 1.2 |
| Maize | 10.9 | 10.6 | 1.3 |

*Table A 7. Simulated crop yields per cultivar and year for the rain feed land.*

| Crop Yields [tons dry matter ha$^{-1}$] | | | |
|---|---|---|---|
| Crops | Median | Mean | STD |
| Cotton | 3.0 | 2.9 | 0.7 |
| Clover | 9.8 | 9.6 | 0.6 |
| Wheat | 3.9 | 3.6 | 0.9 |
| Barley | 4.0 | 3.9 | 0.9 |
| Maize | 9.5 | 9.2 | 1.5 |

1262