# Peer review of "Regional Assessment and Uncertainty Analysis of Carbon and Nitrogen Balances at"

_Biogeosciences, 2023_

## Author Response (AR1)

**Dear Mrs Trebs,**

**Thank you for the review of our manuscript bg-2023-52 as stated below.**

**Public justification (visible to the public if the article is accepted and published):**
Dear authors,

Thank you for submitting your manuscript "Regional Assessment and Uncertainty Analysis of Carbon and Nitrogen Balances at cropland scale using the ecosystem model LandscapeDNDC" to Biogeosciences. After reviewing the comments made by the referees and your response letters, I find that your manuscript requires major revisions to address the main concerns and critical comments of the referees. The quality of the paper is currently not sufficient for publication in BG. The interpretation & discussion of the results and disussion of uncertainty should be improved subtantially. Especially the concerns of Referee #2 should be taken into account. In your revision, please especially address this and also incorporate your replies to the referees. The manuscript will be re-reviewed by at least one referee.

Reviewer 1 (Emanuele Lugato)

Dear editor

The paper of Sifounakis et al., presents a regional simulation of C and N budget in Thessaly (Greece), with the widely used biogeochemical model landscapeDNDC. The strength of the paper is that is driven by detailed regional information and uses an uncertainty analysis based on the Markov Chain Monte Carlo (MCMC) Metropolis–Hastings algorithm. Information about model implementation and parameterization, as well as uncertainty calculations, are quite useful for the GHG inventory community (and beyond), considering the attempt to move toward tier 3 approaches under the EU LULUCF regulation.

Up to the discussion, the paper is substantially well developed and the overall N fluxes (and C to a lesser extent) look reliable according to other studies and my personal experience.

I have, anyway, a series of questions and concerns that, I hope, can lead to some improvements of the manuscript.

- Methods

The European Soil Database (ESDB v2.0, 2004), used as input soil dataset, is quite old indeed. New gridded value are available at ESDAC or alternative at ISRIC (soil grid), which authors may think to consider for the next studies (if not for this).

Answer: This is indeed an important criticism about the regional input data used in the study. As most regional inventory studies consume 75% of the study effort in collecting, aggregating and preparing of regional input data for model initialization and to drive the model along the simulation time span. As the arable land management (crop rotations, residue management and fertilization/manuring) poses the biggest uncertainty for the N balance, we have focused to use the available resources in this study to use most recent regional information of arable

land management for the region, and use an existing model initialization dataset from the EU project NitroEurope based on the European Soil Database (ESDB v2.0, 2004). To our understanding, the improvements in using a newer version of the Soil Database or using ISRIC soil grid data, would improve the accuracy of the inventory calculation to a lower extend than improvements in the description of the real arable land cultivation would have on the carbon and nitrogen cycling of the region. Nevertheless, we aim to follow this comment and to test the influence of different sources of soil data for the simulation of the arable land carbon and nitrogen cycling.

While the uncertainty is an important and valuable part of the work, some clarifications and details are needed, considering that Authors refer to a previous work 'Santabarbara (2019)' missing in the reference list. In the text is stated: "In the current analysis, 500 joint parameter sets were sampled from the posterior distributions in combination with input data perturbations as reported by Santabarbara (2019)".

Answer: Oh sorry for the missing reference and the invalid citation. We have added the reference of Santabarbara (2019) to the manuscript. We use the Mendeley citation software and we will check all citations and references again.

Is not totally clear if MCMC Metropolis–Hastings is performed for this study or if posterior distributions to generate the 500 joint drawing (runs) are derived from a previous work. I also assume that posterior distributions for the initial soil conditions and drivers are generated for each of 1000 polygons simulated (in total I count 1000*10*500 simulations). Moreover, the 24 most sensitive process parameters to gaseous N fluxes are not mentioned and would be interesting to report them in the supplementary.

Answer: We have added a paragraph starting in line 298 explaining the methodology in more details: In a previous study by Santabarbara (2019), an extensive sensitivity analysis on all soil bio-geochemical process parameters, soil initial data and arable management data was performed identifying the 24 most sensitive process parameters (listed in supplementary material), the most sensitive soil initial data (soil profile data on bulk density, soil organic carbon content, pH value) and the most sensitive management information (fertilization and manure N rates, tilling depth) to aquatic and gaseous N fluxes from arable soils. This was digested in the MCMC simulation sampling a combination of 24 parameter values, 3 values of soil initial data and 3 management information. The sampling of the soil initial data as well as the management data was performed as perturbations to the existing data: For each quantity a perturbation was sampled individually and applied to all corresponding values in the soil profile or to all years in the management description. The MCMC simulation performed by Santabarbara (2019) simulated more than 100 000 iterations for various arable sites until the MCMC simulation converged towards a stable combined posterior distribution of parameter values and soil and management input data perturbations. In the current analysis, we have sampled 500 joint parameter / input data perturbation sets from the posterior distributions as reported by Santabarbara (2019) and we deployed them in simulations (propagation through the model) for the regional inventory leading to 500 inventory simulations.

I generally recommend to improve this section providing more details, considering eventually a flowchart to facilitate the reader.

Answer: In the preparation of the manuscript, we aimed to focus on the deployment of the uncertainty analysis rather than details on the methodology itself. We appreciate your comment and will add a section in the supplementary material describing the details of the i) sensitivity analysis, ii) the theory and application of the MCMC simulation to generate the posterior distributions of parameter / disturbance sets and iii) the propagation of the posterior distributions through the model leading to result distributions. But we think that the manuscript should focus on reporting the uncertainties rather than the method.

If I well understood, each crop rotation (R1-R5) is simulated in 100% of the agricultural land and then each crop, in a given year, is weighted based on statistics about corresponding cultivated area. Therefore, for example (table 1-2), the clover increases from 0.15 to 0.39 from 2012 to 2013, but the additional area is implicitly coming totally from a preceding rotation with winter wheat. That may lead to some approximations that can be acceptable, but could be also mentioned.

Answer: Yes, your understanding of the methodology is correct. It is well known from literature, that the agricultural performance as well as the strength of the soil carbon and nitrogen cycling strongly depends on the interaction with crop rotations. Crop rotations will return various nutrients in different quality to the soil and they are key to prevent / interrupt pest and disease cycles. While the former is not implemented in LandscapeDNDC, effects of crop rotations to improve soil health by filling the soil organic matter pools from root litter and above ground residues from different crops (depending on the different litter carbon to nitrogen ratios to be transferred the various soil carbon pools) are well represented in the model (see Haas et al., 2022 (STOTEN) Long term impact of residue management on soil organic carbon stocks and nitrous oxide emissions from European croplands, https://doi.org/10.1016/j.scitotenv.2022.154932). The optimal solution would be to use spatial high resolution detailed information (such as EU invekos data) on field scale to take rotation effects into account. Data protection rules and data availability constraints prevents this such that modelling efforts need to simplify. In our opinion, it we try with the construction of the five rotations to come close as possible to reality while keeping complexity low as possible. In contrary to our very complex approach, recent global and continental inventory simulations such as Jägermeyr, J., et al, 2021 (Climate impacts on global agriculture emerge earlier in new generation of climate and crop models. Nature Food, 2, no. 11, 873-885, doi:10.1038/s43016-021-00400-y. ) perform several single crop monoculture simulations over long time spans. They account only for crop residues from the same crop from previous years repetitively. These approaches are highly vulnerable to artificial soil carbon and nitrogen depletions and accumulations.

We will add a paragraph explaining in more details the construction of the crop rotations in the supplementary material.

- Carbon budget

In general, I found some semantic problems in C budget nomenclature. GPP+ manure cannot be called 'C input' to soil as half of GPP is respired autotrophically and a consistent part removed from field. Moreover, what is called carbon fluxes from the soil to the atmosphere (TER) is properly the 'ecosystems respiration', since it includes both autotrophic and heterotrophic respiration. The terminology should be carefully revised throughout the text.

Answer: The carbon balancing within our study use carbon inputs and carbon outputs from the soil / vegetation system. The carbon inputs into the system are GPP and C input via manure. The C output fluxes are TER and C removed by harvest. We do not distinguish between autotrophic and heterotrophic respiration but as TER combining both. Assuming there were no plants left on the field by the end of the year, our view of the system is correct: (GPP + C input by manure) – (TER and C removed by harvest) = changes in SOC.

The C budget shows an average SOC sequestration rate of 0.5 t/ha per year, which seems a quite high rate of accumulation to my experience. I'm wondering how this number is affected by the model initialization in term of i) uncertainty of initial values, especially if coming from ESDB and ii) SOC pools partition, which is not mentioned/described. Not having a long-term spin-up, the model could be far from its equilibrium state, leading to spurious trends.

The selected soil bio-geochemistry module in the LandscapeDNDC model works in contrast to e.g. the Daycent model different: i) The initial soil carbon value per soil layer in the input file is divided during initialization into the internal 4 carbon pools (depending on their decomposability). This process was well calibrated in the past. ii) To ensure stability with respect to soc input data, the model adjusts the decomposition rates of the 3 soil carbon litter pools (within valid ranges) within the first 3 simulation years. This method has proven to ensure SOC stability and an equilibrium when simulating forest, grassland and arable systems. The model has been extensively tested agains the long term data from the Rothhamsted soil carbon experiments and a dataset from the Askov site in Denmark and a manuscript about details on soc stability and model equilibrium is under preparation.

The explanation that this SOC accrual is 'mainly caused by the inclusion of legume feed crops within the crop rotation leading to increased litter production' is not convincing, unless you assume that those fodder crops were not in the rotation before 2009.

Answer: The C sequestration rate of approx.. 0.5 t/ha per year reported in this study is higher than averages across literature for pure cropping systems. In our study the effect of the perennial legume feed crop within the rotation (which is on the field for 18 months) builds up vegetative carbon stored in roots of several tons per hectare. This carbon will be transferred into the SOC litter pools during the final harvest and tillage. The C sequestration without this grass effect diminish and overall be very depending on the residues management assumed. We have used residues return numbers from a recent study by Haas. et al. (2022) analyzing the effects of SOC dynamics under different residue management scenarios across EU-27.

Here we are faced with the overall uncertainty issue as the data sources differ. Yes it is right, that our soil initialization will most likely not recognize the legume crop shares from the local cropping statistics. Adjusting soil initial data was not an option as this will for sure introduce uncertainty.

- Figures

The quality and details of the figures should be checked with care. For instance, Fig.3 is repeated 3 time without sub-figure labeling (a,b) and, in Fig.8 , the legend is covering the lines.  The Fig. 4 is not so appealing and, maybe, redundant having Fig. 5.

Answer: The issue with the Fig. 3 was created when uploading the file and has been corrected. We will carefully revise the illustrations, captions and sub-figure labeling.

Since the authors made a regional model application, I would have been interested to see some maps, rather than eventually temporal trends that could be accommodated in supplementary material.

Answer: We are well aware of this point. As the study is already quite extensive (regional inventory simulation, presentation of the full C and N balance and presentation of the model uncertainty) we had to limit the analysis for this paper. For the authors, it has been a priority to report the full C and N balance with all fluxes. As there was only one modelling paper found reporting the full N balance (of a region) we are aware that many questions about the robustness of the modelling effort will arise and therefore we decided to report the uncertainty analysis. A detailed reporting and analysis of spatial results (spatial patterns of results and geospatial analysis of soil and climate drivers) would cover a substantial part of the paper, which we will address in a follow up paper for sure.

- Discussion

In my opinion, this is the part of the paper that can be improved more. Most of time, the discussion is developing around the comparison of simulated DNDC fluxes with other modelling or empirical-driven approaches done at different scale, time, in other continents or sites with different pedo-climatic conditions. We all know how fluxes, especially N, are dependent on the local context, therefore, I would only keep the comparisons with data on the specific region.  Eventually, all the other data reported by different studies for the different fluxes could be summarized in a graphical way, shortening the discussion and indicating the limits mentioned above.

Answer: We are aware that reporting N balances for a specific region is a challenge especially when no comparable studies neither experimental nor numerical studies are available. For Greece or the Mediterranean region, we did not find any comparable results. Neither did we find comparable literature reporting at least parts of the N cycle. Our objective was to present the N balance as a new standard when simulating and analyzing any part of the N cycle as requested in a very recent opinion paper by a group of leading scientists working on the modelling of the N cycle in arable systems. (https://www.authorea.com/users/620451/articles/644452-modeling-denitrification-can-we-report-what-we-don-t-know). The paper encourages scientists to report the full N balance

including all subfluxes from any modelling study to the scientific community even though some components have not been validated against observations or remain even questionable. The paper states as conclustion: "We assume that the scarcity of complete (i.e. including N2 fluxes and other N pools/pathways) modeled N balances in the soil denitrification literature stems from the reluctance of the scientific community to support the publication of unvalidated modeled output, especially given that the simulation results of these 'neglected' N pools may be unrealistic. But this self-censorship of authors has resulted in a missed opportunity to share knowledge and improve our understanding of modeled processes. We recommend that future studies exercise transparency in publishing model outputs. We ask authors to focus on the aspects of their model that were of particular interest (i.e. validated model developments), but, while clearly stating which variables were not validated by measurements, to include all related pools and parameters to the fullest extent possible (e.g. all modeled N pools/pathways, soil aeration and CO2 flux). Presenting such results does put additional pressure on the authors, as the presented model outputs have to be sufficiently robust and coherent for publication. However, the publication of the modeled N-balance simulations is crucial for future model development; it would fundamentally improve the robustness of models, speed up fine-tuning and ultimately advance our understanding of the N cycle."

We have outlined this study long time before the paper of Grosz et al. has been written and we tried to focus in our study to present the full N balance including all different N sub fluxes of the underlying system. As this criticism was mentioned by the reviewer #2 as well, we have rearranged the discussion in our manuscript. First we point out the importance of the requests by Grosz et al in our discussion. Then we start with the discussion of the full N balance and have rewritten the subsection to discuss the partitioning of the various N outfluxes in our study. We point out that our study compares well with the only other simulation study found (using LandscapeDNDC as well) analysing the full N balance. We discuss the differences or our modelling approach to other existing approaches using different methodologies and / or a reduces view of the N balance. Finally we discuss the agreement of our modelled N fluxes in comparison to studies reporting simulations of only parts of the N cycle. As no comparable studies for the Mediterranean systems were available, we have compared N fluxes belong to different arable systems, e.g. Austria, Switzerland or Saxony.

I would have expected to understand more about the model behavior in space, which (and why) modifications have been introduced in this run and more about model sensitivity to the different parameters. For instance, a policymaker might be interested tp understand which input information (soil, management, crop etc.) should be improved more to reduce the uncertainty or, if uncertainty has the same magnitude regionally.

Answer: This is a very important issue especially as more and more stakeholders as for easy answers for mitigation. We will aim to address this in a follow up study where we will focus on geospatial result analysis and try to include the mitigation option as well. But due to limitation in space, this can not be addressed for the full N balance in this paper.

I think the paper has good basis, with a consistent model parameterization based on regional data and reliable simulated fluxes, especially for the N cycle. General improvements are needed, especially in the uncertainty methodology description and the discussion, not very informative and comparing often pears and apples.

Comments of Reviewer #2

The spin-up to achieve equilibrium is very short and the SOC results in figure 8 let assume that the system might be not in equilibrium (the changes are not explained in the discussion). I am also wondering about the assumptions made for the residue management, as this will be a crucial impact on the SOC changes. While this aspect was discussed for other studies, there is no analysis for the here presented results.

Answer: The selected soil bio-geochemistry module in the LandscapeDNDC model works in contrast to e.g. the Daycent model different: i) The initial soil carbon value per soil layer in the input file is divided during initialization into the internal 4 carbon pools (depending on their decomposability). This process was well calibrated in the past. Ii) To ensure stability with respect to soc input data, the model adjusts the decomposition rates of the 3 soil carbon litter pools (within valid ranges) within the first 3 simulation years. This method has proven to ensure SOC stability and an equilibrium when simulating forest, grassland and arable systems. The model has been extensively tested against the long term data from the Rothamsted soil carbon experiments and a dataset from the Askov site in Denmark and a manuscript about details on soc stability and model equilibrium is under preparation.

The C sequestration rate of approx. 0.5 t/ha per year reported in this study is higher than averages across literature for pure cropping systems. In our study the effect of the perennial legume feed crop within the rotation (which is on the field for 18 months) builds up vegetative carbon stored especially in roots of several tons per hectare. This carbon will be transferred into the SOC litter pools during the final harvest and tillage. The C sequestration without this grass effect diminish and overall be very depending on the residues management assumed. We have used residues return numbers from a recent study by Haas. et al. (2022) analyzing the effects of SOC dynamics under different residue management scenarios across EU-27. Residues return values were selected as in the "Basecase" scenario for Haas at all (2022) showing a stable SOC dynamics for a 3 model ensemble.

I am not sure, why climate change mitigation is listed in the keywords. This is not part of this study. Actually, this study missed the opportunity to analyse the different fluxes against each other, as there are not many studies analysing bot: C- and N-fluxes. This provides the opportunity to analyse, if SOC gains will be compensated by N-emissions. Is the spatial distribution relevant for this compensation? ….

Answer: The use of mitigation is unfortune as the study is nor focusing on mitigation. The keyword was used to indicate the results to be used as a baseline for the N balance. We will correct this.

We are well aware that we did not analyse any process details leading to the different N fluxes in the modelling study. As the study is already quite extensive (regional inventory simulation, presentation of the full C and N balance and presentation of the model uncertainty) we had to limit the analysis for this paper. For the authors, it has been a priority to report the full C and N balance with all fluxes as requested by Grosz et al. (2023). As there was only one modelling paper found reporting the full N balance (of a region) we are aware that many questions about the robustness of the modelling effort will arise and therefore we decided to report the results including an extensive uncertainty analysis.

A detailed reporting and analysis of spatial results (spatial patterns of results and geospatial analysis of soil and climate drivers) would cover a substantial part of the paper, which we will address in a follow up paper for sure.

I suggest to analyse the own results more detailed and provide an appropriate discussion of the results.

Our objective in this study was to present the N balance as a new standard when simulating and analyzing any part of the N cycle as requested in a very recent opinion paper by a group of leading scientists working on the modelling of the N cycle in arable systems. (https://www.authorea.com/users/620451/articles/644452-modeling-denitrification-can-we-report-what-we-don-t-know). The paper encourages scientists to report the full N balance including all subfluxes from any modelling study to the scientific community even though some components have not been validated against observations or remain even questionable. The paper states as conclusion: "We assume that the scarcity of complete (i.e. including N2 fluxes and other N pools/pathways) modeled N balances in the soil denitrification literature stems from the reluctance of the scientific community to support the publication of unvalidated modeled output, especially given that the simulation results of these 'neglected' N pools may be unrealistic. But this self-censorship of authors has resulted in a missed opportunity to share knowledge and improve our understanding of modeled processes. We recommend that future studies exercise transparency in publishing model outputs. We ask authors to focus on the aspects of their model that were of particular interest (i.e. validated model developments), but, while clearly stating which variables were not validated by measurements, to include all related pools and parameters to the fullest extent possible (e.g. all modeled N pools/pathways, soil aeration and CO2 flux). Presenting such results does put additional pressure on the authors, as the presented model outputs have to be sufficiently robust and coherent for publication. However, the publication of the modeled N-balance simulations is crucial for future model development; it would fundamentally improve the robustness of models, speed up fine-tuning and ultimately advance our understanding of the N cycle."

We have outlined this study long time before the paper of Grosz et al. has been written and we tried to focus in our study to present the full N balance including all different N sub fluxes of the underlying system. As this criticism was mentioned by the reviewer #2 as well, we have

rearranged the discussion in our manuscript. First we point out the importance of the requests by Grosz et al in our discussion. Then we start with the discussion of the full N balance and have rewritten the subsection to discuss the partitioning of the various N outfluxes in our study. We point out that our study compares well with the only other simulation study found (using LandscapeDNDC as well) analysing the full N balance. We discuss the differences or our modelling approach to other existing approaches using different methodologies and / or a reduces view of the N balance. Finally we discuss the agreement of our modelled N fluxes in comparison to studies reporting simulations of only parts of the N cycle. As no comparable studies for the Mediterranean systems were available, we have compared N fluxes belong to different arable systems, e.g. Austria, Switzerland or Saxony.

The conclusion of the paper was to motivate the modelling community to report the overall N balance and not only sub fluxes even some sub results may be assigned with high uncertainties. The community needs these results to identify shortcomings and derive decisions where to focus model adaptations / developments.

Some additional comments:

Abstract

There are no comments or results on the uncertainty analysis in the abstract. Lines 24-34 are only a list of results, but do not summarise the paper or aggregate the outcome of the study.

Answer: We report in the abstract the mean values for all components of the N outfluxes with their uncertainty ranges given in brackets. We have added this to the abstract.

Graphical abstract

The picture is nice, but it shows only the nitrogen fluxes, but not any carbon fluxes or any information about the uncertainty analysis. This means, that this does not summarise the study.

Answer: According to the opinion paper of Grosz et al. (2023), the novelty and therefore the focus of our study was on the presentation of the N balance and all the N outfluxes. The C budget has been extensively reported by other studies before.

Methods

Table 1 indicates corn as maize, summarising food corn and silage maize. For modelling, however, there is a crucial difference, as the residue treatment differs and will affect large differences for the carbon input to the soil.

Answer: Yes this is correct and has been a problem in the study as well. Unfortunately, the statistics for the region do not distinguishes between food corn and silage maize. We are aware that especially for the resiudes management this is crucial. We did mention this in the paper and will extend it in the supplementary material explaining the crop rotations in details.

Table 2 suggest that the rotation changed, which does not make sense. Does this represent the crop coverage for the corresponding year or is a new rotation introduced each year? I assume this is due to the fact that there is a five year rotation applied on a 8 year simulation period with different representations. Is this possibly affecting the results?

Answer: It is well known from literature, that the agricultural performance as well as the strength of the soil carbon and nitrogen cycling strongly depends on the interaction with crop rotations. Crop rotations will return various nutrients in different quality to the soil and they are key to prevent / interrupt pest and disease cycles. While the former is not implemented in LandscapeDNDC, effects of crop rotations to improve soil health by filling the soil organic matter pools from root litter and above ground residues from different crops (depending on the different litter carbon to nitrogen ratios to be transferred the various soil carbon pools) are well represented in the model (see Haas et al., 2022 (STOTEN). Long term impact of residue management on soil organic carbon stocks and nitrous oxide emissions from European croplands, https://doi.org/10.1016/j.scitotenv.2022.154932).

We did not have any detailed information on crop rotations for the region. The only data we could derive were crop cultivation statistics. Therefore we had to derive a suitable crop rotation. At the same time, we had data of the share of arable land used per crop in the region. Therefore we duplicated the crop rotation in such a way (shifted them by one year) that each crop occurs only once per year in one of the five rotations. Then the final five rotations were extrapolated into the future.

The optimal solution would be to use spatial high resolution detailed information (such as EU invekos data) on field scale to take rotation effects into account. Data protection rules and data availability constraints prevents this such that modelling efforts need to simplify. In our opinion, it we try with the construction of the five rotations to come close as possible to reality while keeping complexity low as possible. In contrary to our very complex approach, recent global and continental inventory simulations such as Jägermeyr, J., et al, 2021 (Climate impacts on global agriculture emerge earlier in new generation of climate and crop models. Nature Food, 2, no. 11, 873-885, doi:10.1038/s43016-021-00400-y. ) perform several single crop monoculture simulations over long time spans. They account only for crop residues from the same crop from previous years repetitively. These approaches are highly vulnerable to artificial soil carbon and nitrogen depletions and accumulations.

Yes crop rotations will influence overall results, but we have checked with local farmer advisers for the most suitable rotation out of our 5 crops. Changing the sequence of the different crops within the rotation and average across the five rotations will only change the results marginally and by orders of magnitude less than e.g. the variation in process parameters, soil initialization and management data. Therefore this can be neglected.

In the supplementary material, we will add a paragraph explaining in more details the construction of the crop rotations and their interactions.

Details about the management are missing. How are the dates for the different agricultural management (sowing, tillage, fertilizer application, etc.) options are derived? Are they constant or dynamic? It is referred to farmers knowledge, but it would be good to get an idea about temporal and spatial variability.

Answer: The crop rotation was static for all polygons. The system has been tested before for single polygons. Timings and management details have been derived from local farmer advisers. We will explain this in details in the supplementary material section.

Why is the yield not evaluated annually?

Answer: We aimed with the study to address the presentation of the full N budget of a cropping system. The reporting and evaluation of the yield is only to conclude on the model's performance against the only available validation data. Again, the uncertainty in the reported data is high as e.g. data for food corn and silage corn is aggregated into one category…

Finally we want to state that in Germany there has been a six years research initiative about Denitrification just recently ended with an opinion paper (Grosz et al. 2023) concluding that the scientific community still lacks insights into the overall N budgets in agricultural systems and their representation in the different process based models. Only very few recent papers started reporting and analyzing the overall N balance in modelling and experiments, especially using stable isotope measurements to address denitrification and N2 losses.

But for the modelling community to our knowledge, any model targeting N2O would always report the overall N budget. The Denitrification initiative concluded that there were no N budgets reported in the past such that conclusions on the performance of the different models could not be derived. (Except the only study by Schroeck using the LandscapeDNDC model). The conclusion of the paper was to motivate the modelling community to report the overall N balance and not only sub fluxes event though hardly no comparable studies were available for comparison.

---

## Author Response (AR2)

BG-2023-52
*Regional Assessment and Uncertainty Analysis of Carbon and Nitrogen Balances at cropland scale using the ecosystem model LandscapeDNDC*
Odysseas Sifounakis, Edwin Haas , Klaus Butterbach-Bahl , and Maria P. Papadopoulou
Corresponding author: Edwin Haas (edwin.haas@kit.edu)

**Editor comment**
Dear authors,
thank you very much for revising the manuscript. Referee #3 still does not find the manuscript convincing and some of the concerns of the first round of reviews (uncertainty analysis, discussion of results, model calibration/initialization) are still not entirely fixed. The main weakness of the study is the missing comparision to data measured in the field, but I am aware that this cannot be changed. Also, the plausibility of the C sequestration is questioned by the referee, although I note that you have discussed the reasons for this. Please provide a point-by-point response, incorporate changes into the manuscript or provide a strong rebuttal of the comments. Also, in the conclusion you should mention that the model output was not compared with field data, which is a major weakness and therefore, I would not refer to it as "prototype study". Technically, the paper would profit from proofreading by a native English speaker and the quality of the figures should be improved (some of them are hardly readable).
Best regards
Ivonne Trebs

Answer:
Dear Editor,
thank you very much for your effort in the review process of the manuscript.
Compiling a regional inventory of nitrogen fluxes for arable land requires the use of models to upscale to various soil, environmental and climatic conditions. Even with available field observations, modelling would be needed for conditions not incorporated/reflected by the measurements.
For the US greenhouse gases reporting to the UNFCC by the US EPA (https://www.epa.gov/system/files/documents/2022-04/us-ghg-inventory-2022-main-text.pdf) process-based modelling (IPCC Tier III using the DayCent model) on the national scale is deployed for the assessment of $N_2O$ and $CH_4$ emissions from managed agricultural soils. The responsible investigators Stephen Del Grosso (Colorado State University) and Stephen Ogle (US Dep. Agriculture) preformed a similar approach compiling a national inventory in combination with an uncertainty assessment. The report only states that the model is continuously tested on data from long-term field experiments. The same applies to our model (https://ldndc.imk-ifu.kit.edu/map/map.php). The EU urgently needs more regional and national inventory simulations to establish IPCC Tier III methodology for UNFCC reporting for the EU and/or some member states.
Questioning about the soil C sequestration is a good example of why the reporting of all fluxes is beneficial, as it opens the possibility to assess the greenhouse gas balance by comparing C sequestration versus $N_2O$ emissions in $CO_2$ equivalents (instead of the isolated

view on soil C sequestration), leading to a much more diverse picture of the arable management.

As pointed out by you and the reviewer, we have added the statement "Overall, one should be aware that the simulation results were not compared to field data for the region, which is a major weakness of the study." Additionally, we have removed the statement about the paper being a "prototype study".

The manuscript has been reviewed by a native English speaker to improve its readability.

We have replaced all figures with high quality figures in 300 dpi resolution.

In the name of all co-authors, we thank you for your effort in the review and to improve the manuscript.

**Answers to the reviewer questions and statements**
Report #1
Anonymous referee #3

We have numbered the resulting reviewer comments into sections. This helps to refer to previous or later answers to reviewer questions.

1.      This manuscript reports on a study where the landscape DNDC model was used to assess all major C and N fluxes and their uncertainty for a region in Greece. New in this study is the region for which this assessment was done, using the DNDC model. New is also that the authors report all C and N fluxes, which is often not done and which was recently identified as a problem to better understand denitrification in an unreviewed opinion paper (Grosz et al., 2023).

Answer:

The paper (Grosz et al., 2023, https://doi.org/10.1029/2023AV000990) has been published in AGU Advances and has been selected as an editorial highlight in EOS.org. The key message stated is "*that it is crucial to publish and communicate the complete N cycle as calculated by the models. This practice is vital for advancing model development, ensuring quality control, facilitating model intercomparison, and generating new hypotheses for empirical field studies.*"

The paper by Grosz was co-authored by researchers such as Klaus Butterbach-Bahl, Eric Davidson and Steve Del Grosso which are among the most cited scientists working on (modelling) the nitrogen cycle and it emerged out of a scientific meeting on modelling denitrification. The conclusion reached at that meeting was, that there is only one modelling paper published reporting the full nitrogen cycle so far. There is a large scientific knowledge gap on the established and commonly used ecosystem models and their capabilities in terms of modelling denitrification. The model developers urgently need such full nitrogen flux studies to compare and evaluate the various implemented concepts and mechanisms representing soil nitrification and denitrification. If we continue to not report all fluxes or put the "unmeasured fluxes" into the appendix at most, this scientific knowledge gap will never diminish or disappear.

Therefore, we strongly defend our approach to report and discuss all carbon and nitrogen fluxes.

2.      Although I did not read it in the manuscript (or I missed it), I assume that the standard calibration of the DNDC model was used. I am not an expert on initialization of models and their uncertainty analyses and assume that one of the other reviews is sufficient knowledgeable to review this aspect, so I am not going to comment on this.

Answer:

Calibration

Yes, the model has been used with the "standard calibration," as reported by Molina et al. (2016, 2017) and mentioned in the Material and Methods section. This calibration was comparing field observations of pant biomass and yield at different vegetative stages, soil $N_2O$ and NO emissions, soil $NO_3$ concentrations and $NO_3$ leaching against simulation results. Our validation datasets (ICOS site Borgo Cioffi is comparable, same latitude and similar altitude 30-250 m a.s.l. as the majority of the Thessaly arable land is on) partly comprise environmental conditions that are similar to the study area / region.

Uncertainty Analysis

Especially when reporting all carbon and nitrogen fluxes it is fundamental to accompany the approach by a full model uncertainty analysis. As stated by Grosz and reported and discussed in the manuscript, some fluxes cannot be validated on field scale at all. The uncertainty analysis reports the possible range of joint model results for the nitrogen balance instead of one specific representation, which creases credibility and trustworthiness. This methodology in reporting model uncertainty is widely used e.g. in hydrology or in climate science.

3.      I am worried about the limited amount of data that the simulated results were compared to, which seems to be mainly crop yield data (Table 3). I am aware that DNDC has been calibrated in other studies, but this does not include the region where this study was done.

Answer:

The main objective of this study was to assess carbon and especially nitrogen fluxes of arable land cultivation for the region. One fundamental purpose of process-based models (as advised by IPCC) is the transfer of measured quantities to other climate and soil conditions, where no measurements and no observations are available for the region, country or for a similar Mediterranean system, process-based modelling (IPCC Tier III) has been selected as the most adequate tool.

In fact, the EU has supported within the 7[th] framework the NitroEurope project to establish the "European nitrogen assessment" in which several observation sites running nitrogen flux measurements have been supported. Unfortunately, none of these sites was located in the Mediterranean region. In literature, only some rare $N_2O$ soil emissions were reported e.g. for a site in south Italy. The station Borgo Cioffi (ICOS code IT-BCi) is an agro-ecosystem, typical buffalo farm growing corn and forage, in Mediterranean conditions. The site is located in the territory of the municipality of Eboli, province of Salerno, Southern Italy, 5 km from the sea (https://meta.icos-cp.eu/resources/stations/ES_IT-BCi). For several years, a chamber measurement system was installed to assess soil-based $N_2O$ emissions from a corn monoculture rotation. This is the only observation site found in literature, which is comparable (same latitude,  most of the Thessaly arable land is on similar altitude 30-250 m a.s.l.) and supplies only $N_2O$ emission observations on a coarse temporal resolution. This site was included in the LandscapeDNDC calibration study by Molina et al. (2016) and is therefore reflected in the general model behavior.

4.      Whereas I do understand the argument that all fluxes of a modelling exercise should be reported, we should also not forget that these fluxes are unvalidated (l.91) and some of these fluxes may be unrealistic (l. 92).

Answer:

The recommendation from Grosz et al., (2023) was a direct result of a meeting on modelling denitrification in agroecosystems. Measurement techniques for denitrification (after the identification of the deficit of the acetylene inhibition method) have advanced, such as helium incubation systems and $^{15}$N isotope measurements. None of the available bio-geochemical models is capable of simulating $^{15}$N isotope abundance. The models are not validated for $N_2$ emissions at all. As long as we do not report such fluxes the model developers cannot intercompare their process descriptions, model behavior etc. and this makes it very complicated to advance/improve and validate the process description of denitrification. We do not even know whether simulated $N_2$ fluxes are "unrealistic" because none of these $N_2$ fluxes were published in the past.

We believe, that science is obliged to close this knowledge gap, especially as some of these models were used for UNFCC greenhouse gas reporting.

5.      For outsiders the uncertainty analyses will make the results more believable, but let's not forget that an uncertainty analyses does not replace measurements. So where does this leave us?

Answer:

We agree in general, that it would be good to compare and validate simulated results to observations. But what if there are no such observations (which is the case for our region and most parts of the world)?

A calibrated model, which has proven its ability to predict a single or combined nitrogen fluxes, is - according to IPCC - the recommended tool (Tier III) to assess nitrogen fluxes in agroecosystems and is associated with the lowest uncertainties. It is more reliable than the Tier I and II approaches and in combination with an extensive uncertainty analysis, the preferred approach to gain insights.

On the other hand, only a small number of temporal high-resolution soil $N_2O$ emissions measurements exist; many $N_2O$ emission measurements were only conducted during the cultivation season of fertilization experiments, neglecting the rest of the year where e.g. crop and root residues decomposition could lead to major $N_2O$ pulses not being considered in observations (Olesen et al. 2023, GCB, https://doi.org/10.1111/gcb.16962). Another source of uncertainty according to Wagner-Riddle et al. (2017, Nature, https://doi.org/10.1038/ngeo2907) are freeze-thaw cycles in arable soils, which have been largely neglected in the past in soil $N_2O$ measurement campaigns. The scientific community did not provide enough suitable observations to serve as a foundation for validating all modelling approaches.

Assuming we can use available and suitable soil $N_2O$ emissions to validate $N_2O$ modelling results, two studies report $N_2O$ emissions with the same quality ($r^2$, RMSE, NRMSE, BIAS, NE etc.). Why should we believe the study reporting only soil $N_2O$ emissions more than a study reporting all nitrogen fluxes? Comparing all other nitrogen fluxes with common knowledge of the underlying system will help to increase the reliability of the study, as any additional nitrogen flux which is in agreement will constrain the nitrogen cycle. On the other hand, a model reporting only $N_2O$ emissions could be completely overparameterized providing matching soil $N_2O$ emissions while other nitrogen fluxes are outside possible valid ranges. We

know, for example, from experiments for denitrification plausible $N_2/N_2O$ ratios for different agricultural practices. Reporting of $N_2$ emissions would help to estimate the contribution of denitrification on any modelled $N_2O$ emission.

So where does this leave us?

- Models are tools to assess nitrogen fluxes when no measurements are available
- Uncertainty analysis needs to accomplish reporting of all carbon and nitrogen fluxes
- Reporting all fluxes can prevent from drawing misleading conclusions
- Comparing all nitrogen fluxes with common knowledge increases reliability
- Considering additional nitrogen fluxes will gain more insights into the system
- Intercomparing all nitrogen fluxes between models will improve model development
- Reporting all nitrogen fluxes will widen the scientific audience of a reported study and make it more useful
- For the sake of completeness, we produce it and therefore we should report it
- Personally, I do not see any disadvantage in reporting it with the uncertainty analysis

6.       I think reporting of fluxes and uncertainty should be done, but it could also be done in an appendix. Right now, reporting all fluxes appears to be the major goal of this study and is prominently discussed. I am wondering how the outcome, and the reporting of all fluxes in this modelling exercise will be used?

Answer:

The study of the soil nitrogen cycle is a fundamental objective of soil bio-geochemistry, as it determines soil health and fertility as well as all fluxes of excess nitrogen into the environment, in case of fertilized systems. As a model developer, the representation of the soil nitrogen cascade is the fundamental objective while we and other users of the models have tended in the past to only report soil carbon dynamics and/or $N_2O$ emissions. If we do not trust the representation of intermediate components of nitrogen cascade and especially all other nitrogen fluxes out of the system, how can we trust the $N_2O$ emissions? According to Grosz, some of these fluxes may be simulated with uncertainty, but we do not know. We cannot leave this scientific knowledge gap unresolved, as the use of these models becomes more and more important in e.g., carbon farming, assessments of carbon credits in agriculture, UNFCC greenhouse gas reporting etc.

As illustrated in the answer below, the combined reporting of soil C stock changes and soil $N_2O$ emissions in terms of the greenhouse gas balance (converting $N_2O$ emissions into $CO_2$ equivalents offsetting soil carbon sequestration) is superior to concluding soil carbon changes only considering SOC dynamics.

7.       If this manuscript will be published a policy maker could use it to argue that current management in the study region leads to soil C sequestration of 0.5 tons C ha-1 yr-1, and so farmer should get carbon credits for their environmentally friendly management. Really? Is there any study based on field data, that actually shows and increase in SOC for this region? Last time I checked, agricultural soils in the EU were losing carbon when they were managed in a conventional way, and I am not aware that this is different in Greece. I think this illustrates the main problem the way this manuscript is written.

Answer:

We respect your criticism on the results of the simulated soil C sequestration in this study, but your criticism highlights why the scientific community should never focus on study results being solely based on one objective / outcome, in this case, the soil carbon. If you do

so, according to your criticism, a policymaker could draw this conclusion. Instead, if you would consider our holistic approach, you can easily assess the climate neutrality of the underlying management, based on considering the total greenhouse gas balance in $CO_2$-equivalents instead of the soil C view. The $CO_2$-equivalents would include soil-based $N_2O$ emissions due to fertilization, which will likely compensate for the soil C gain.

As discussed with the previous reviewers, the rotations used in this study consist of the 5 main crops in the Thessaly region. One reported crop is a legume feed crop. This feed crop is typically sown in late autumn, cut during the following year and tilled in as green manure in the year after, weeks before the next summer crop is sown. Therefore, the perennial feed crop is on the field for about 18 months and develops a large proportion of below ground root biomass which will be tilled into the soil, adding up to 2-3 tons of carbon per hectare for this year. As we have no data on the crop cultivation of each polygon available, we need to assume a best guess rotation, given in Table 1. We have data on which crop was cultivated on which portion of the total arable land per year, such that we scale each rotation for each year by this factor / percentage when aggregating to the total of the years. The total result for each polygon for every year is the weighted sum of five different simulation results, each considering a different crop for the year. This means that the clover feed crop contributes between 15 (2012) to 43 % (2015) of the aggregated result for each polygon. For the soil carbon, we, therefore, simulate a mean across all polygons and years of about 0.5 t C / ha to be realistic, as it includes a large proportion/share of grassland incorporation. The average carbon sequestration of 0.5 t C / ha compares well to results in Haas et al. (2023), where the effect of different proportions of crop residues left on the field have been compared to soil carbon changes, and the associated overall greenhouse gas balance expressed in $CO_2$ equivalents. This has been pointed out by one of the previous reviewers and therefore discussed extensively in the manuscript. Therefore, we aim to keep Table 1 and 2 in the manuscript to improve the understanding of the design of the crop rotations and their aggregation.

The approach is commonly used in regional modelling (Haas et al. 2013, Molina et al. 2017), but in our case, it shows high variability in the reported agricultural cultivation statistics, which was discussed in the manuscript as well.

Overall, we strongly believe, that our approach supplies more opportunities to the scientific community than most studies presented in the past. Other readers might be interested in e.g., the nitrate leaching of excess nitrogen into surface waters or into emissions of reactive nitrogen to the atmosphere after slurry applications, as ammonia. We strongly believe that the scientific audience of many modelling papers could be increased when presenting the total carbon and nitrogen balances and this would largely benefit the scientific community.

8.      The authors have taken the recommendation from Grosz et al., 2023, reported all fluxes and try in their discussion to make the numbers plausible by citing studies conducted in other regions with different soils and different climates. I think this is not really improving the situation compared to not reporting the fluxes as is currently done.

Answer:

We are aware that the paradigm of reporting modelling results which have not been directly validated by observations will cause controversial discussions, otherwise, the paper of Grosz et al., (2023) would have been obsolete.

So why does the scientific community not supply sufficient observation data to fully validate ecosystem bio-geochemical models? Because it is very difficult or even impossible to conduct such observation at a field scale. $NO_3$ leaching observations need disturbances of the soil and

typically collect water over a time period, such that measurements are not temporarily high resolution but integrals over some time. Estimates on field scale Ammonia volatilization/ emission are typically done by acid traps in combination with micrometeorology. Again, they supply integral observations over the sampling time. Full $NH_3$ EC measurements are still very rare. Estimates on the field scale denitrification strengths, including $N_2$ emission measurements, are still impossible and need the use of $^{15}N$ stable isotope measurements. Denitrification can be estimated in incubation systems e.g., under a helium environment. The process understanding behind any of these measurements can be improved by the use of models, as the models always include the full nitrogen cycle at any given point in time. There is no discussion on the usability of the models for $N_2O$ emissions

9.      It would make more sense to report fluxes (maybe in the appendix), so that any scientist who wants to compare his measured or modelled fluxes to the ones reported in this study, while acknowledging that these fluxes have not been compared to any measured data. This is presently not done and I think that is a major problem with this study.
Answer:
In the manuscript we have discussed all fluxes in detail, which has even led to some criticism. This discussion reveals the modelling quality of each component of the carbon and nitrogen fluxes presented. The overall "validation" of this modelling study must be seen as the sum of the various individual evaluations and comparisons of the different components, which we have done extensively in the discussion. As there was only one study reporting all nitrogen fluxes, we have to discuss a comparison to this study even under the constraint that climatic and soil properties differ. Other studies reported more than one flux, others included an uncertainty analysis. For readers using other process-based models not being interested in observations, this would be at least as important as measurements; therefore, should we mark the modelled fluxes with different quality indicators resulting from comparison to other modelling studies?
We assume that the overall "validation" of this modelling study is supported by the large number of components which compare well with everything reported in the scientific literature. In other words, when all nitrogen fluxes, except one, compare well with results in literature, then this one remaining flux, namely the $N_2$ flux from denitrification (for which no measurement results are available on a field scale for normal conditions) is highly constrained by the underlying processes and the mass conservation and therefore cannot be completely off.
We do not feel comfortable marking parts of the nitrogen fluxes in e.g. in Figure 4 and Figure 5 with markers determining their very subjective quality of comparison with observation data. First, this is not (and has never been intended to be) an evaluation study aiming to compare modelling versus observations. Second, the determination of whether a flux has been validated with measurements cannot be objective as such observations do not exist, not for the region and not for any system so far. The fact that such observations do not exist, should never be a reason not to publish such a modelling work as our study. Concluding, we have the certainty that there will be an interest in researchers to read about estimated flux values as this is of scientific importance.

10.     The manuscript is very long and definitely would be more readable if it was further condensed. The discussion encompasses 13 pages! You lost me after page 4 or 5, and I think I will not be the only reader.

Answer:

This is a valid point and we have tried to shorten the discussion and focus on the main points.

11.     Throughout the manuscript, use of English language should be improved. For example, there appears to be unnecessary use of the word 'the' in several parts of the manuscript. I am not going to make more suggestions, since I don't think this is my task as a reviewer, but the manuscript definitely needs copy-editing.

Answer:

Thank you, this is indeed an issue. We have done a review of English and writing at the end of this review process to improve the reading of the manuscript.

12.     Your graphical abstract is not informative at all. First you use several acronyms without explanation. Are these distributions? Then it is unclear what these fluxes are from (a region? A crop? A crop rotation?). The colours are confusing, what do they mean? Green = good and Red = bad?

Answer:

The graphical abstract is the Figure 5 of the manuscript. As explained in the answer to question / comment 17:

Figure 5 is the waterfall diagram of all nitrogen fluxes including the illustration of the uncertainty of each individual flux (illustrated by its distribution). This is an innovative way to illustrate the various nitrogen fluxes as a nitrogen balance. It groups the various fluxes by colors (gaseous fluxes = red, aquatic fluxes = blue, nitrogen export via crop yield and feed harvest = green and all nitrogen input fluxes such as fertilization, manure, biological nitrogen fixation and deposition in brown). The cumulative nature of the waterfall diagram assess the result of the nitrogen balance: a net nitrogen flux into the soil of approx. 13.8 kg-N ha$^{-1}$ yr$^{-1}$. The waterfall diagram is a cumulative illustration of all mean fluxes. Event though it may not be immediately clear to all readers, it becomes clearer as soon as the information in the abstract were considered additionally, which many readers do when searching for information in papers.

13.     Table 1. is not necessary. It just shows that the rotation is followed, for which you don't need a table.

Answer:

We strongly emphasize keeping the table in the manuscript to help in explaining the selection of the crops and construction of the rotations, which were simulated in parallel for each polygon. As this was asked by a reviewer before, we will try to improve the manuscript in this context.

14.     Table 2. I found confusing. If the rotations are followed as shown in Table 1, how is it possible to the contribution of different rotations is changing? Or in other word, if the contribution of rotations is changing, for me this means that the rotations are not followed. Am I missing something? I know that an earlier reviewer also commented on this, but I did not find the answer satisfying. If for some reason it is important to change the contribution

of different rotations, I suggest that you put this in the appendix, but not in a prominent place in the manuscript.

Answer:

This was considered in the answer to question 7. It is important to some readers to fully understand the methodology of how these inventory simulations were constructed as this is not yet a standard procedure. Some high-ranking publications, for example, have been simulating mono-crop rotations for each grid cell (like maize/maize/maize/maize and wheat/wheat/wheat/wheat etc.) and then average across them. They neglect the different portions/shares each crop has (which, in our case, vary with time due to Greek statistics) and the fact that mono-crop rotations have severely different effects, as rotations with sequences of varying crops. We do not estimate the effects of constructing different rotations, which would be a different study.

15.	Figure 1 is not self-explanatory. Acronyms are not explained and colours are confusing

Answer:

We changed the figure caption to improve its understanding.

Figure 1. Distributions of simulated crop yields across all polygons and evaluation period 2012 - 2016 for irrigated and rain feed conditions from the 500 inventory simulations. Colored horizontal lines indicate overall mean, maximum and minimum yields of the distributions, black lines indicate medians of simulated yields. (COTT = cotton, WIWH = winter wheat, WBAR = winter barley, SICO = silage corn, FEED = perennial clover feed crop)

16.	Table 4. This table could easily be abused. Unless you have any field data that supports such a claim, I suggest moving it to the appendix, and adding a clear statement that there are not field data-based studies that support these modelling results.

Answer:

As answered in question 7 above, we would like to present all fluxes in figures and tables. We have added in the table caption a statement that the underlying management and crop rotation include a large proportion of a perennial feed crop incorporation as shown in the C input in Table 4.

We added "Note: The underlying arable management / crop rotations include the ploughing in of a perennial feed crop leading to large C inputs to the soil."

17.	I am not sure if Table 5, Figure 4, Figure 5 and Figure 6 are all necessary and should have such a prominent place in the manuscript. First it would be good to learn in Table 5 which numbers were modelled and which were measured. In addition, the table and these figures basically repeat the same information, which is only presented in different ways. I think Table 5 is enough and could also be moved to the appendix, similar to Table 4.

Answer:

We strongly emphasize to keep the reporting of all carbon and nitrogen fluxes in the paper. Therefore, tables summarizing all data shown in the various figures are a requisite.

Further, the Figures 4 illustrates the proportion of the different nitrogen fluxes in relation to the total input and output flux, which is an important visualization of a fundamental relationship. Readers searching for proportions of various sub-fluxes in relation to input and output fluxes would be keen to see this.

Figure 5 is the waterfall diagram of all nitrogen fluxes including the illustration of the uncertainty of each individual flux. This is the most fundamental diagram as it groups the various fluxes (gaseous = red, aquatic = blue, nitrogen export via crop yield and feed harvest

= green and all nitrogen inputs such as fertilization, manure, biological nitrogen fixation and deposition in brown) and assess the result of the nitrogen balance: a net nitrogen flux into the soil of approx. 13.8 kg-N ha$^{-1}$ yr$^{-1}$. The waterfall diagram is a cumulative illustration of all mean fluxes. Figure 5 illustrates the individual uncertainty of each nitrogen flux.
Figure 6 illustrates the overall uncertainty in the net nitrogen balance. In contrast to Figure 5, here the net nitrogen balance for each of the 500 inventory simulations have been calculated. Then the resulting distribution was illustrated including mean and median. In Figure 5, first the total fluxes were estimated via their means and then these means were summed up cumulative in the waterfall (positive values going to the right, negative values going to the left). The distributions presented in Figure 6 represent a more reliable assessment of the resulting net nitrogen balance and its uncertainty. Such an assessment has not been reported for any system before.

18.      Figures 7 and 8 need improvement. I don't think that the legend needs to be repeated for each panel and in Figure 7 the legend even covers the lines.
Answer:
We will reprocess the subfigures and take care that the legend will be included only one time and will be placed correctly not to cover any data shown in the graphs.